
**Incorporating remote sensing ET into Community Land Model version 4.5**
Dagang Wang[1,2,3,4]*, Guiling Wang[4], Dana T. Parr[4], Weilin Liao[1,2], Youlong Xia[5], Congsheng

4                                              Fu[4]

[1] School of Geography and Planning, Sun Yat-sen University, Guangzhou, China
[2] Guangdong Key Laboratory for Urbanization and Geo-simulation, Sun Yat-sen University,

8                                    Guangzhou, China

[3] Key Laboratory of Water Cycle and Water Security in Southern China of Guangdong High

10                    Education Institute, Sun Yat-sen University, Guangzhou, P.R. China

[4] Department of Civil and Environmental Engineering, University of Connecticut, Storrs, USA
[5] National Centers for Environmental Prediction/Environmental Modeling Center, and I. M.

13                    System Group at NCEP/EMC, College Park, Maryland, USA

*Corresponding author: Dr. Dagang Wang, School of Geography Science and Planning, Sun
Yat-sen University, 135 West Xingang Road, Guangdong Province, P. R. China 510275.
Telephone: (86) 2084114575, Fax: (86) 2084114575, Email: wangdag@mail.sysu.edu.cn





24          **Abstract**

Land surface models bear substantial biases in simulating surface water and energy budgets

despite of the continuous development and improvement of model parameterizations. To reduce
model biases, Parr et al. (2015) proposed a method incorporating satellite-based evapotranspiration
(ET) products into land surface models. Here we apply this method to the Community Land Model
version 4.5 (CLM4.5) and test its performance over the conterminous US (CONUS). We first
calibrate a relationship between the observational ET from the Global Land Evaporation
Amsterdam Model (GLEAM) product and the model ET from CLM4.5, and assume that this
relationship holds beyond the calibration period. During the validation or application period, a
simulation using the default CLM4.5 ("CLM") is conducted first, and its output is combined with
the calibrated observational-vs-model ET relationship to derive a corrected ET; an experiment
("CLMET") is then conducted in which the model-generated ET is overwritten using the corrected
ET. Using the observations of ET, runoff, and soil moisture content as benchmarks, we
demonstrate that CLMET greatly reduces the biases existing in CLM. The improvement differs
with region, being more significant in eastern CONUS than western CONUS, with the most
striking improvement over the southeast CONUS. This regional dependence reflects primarily the
regional dependence in the degree to which the relationship between observational and model ET
remains time-invariant (a fundamental hypothesis of the Parr et al. method). The bias correction
method provides an alternative way to improve the performance of land surface models, which
could lead to more realistic drought evaluations with improved ET and soil moisture estimates.

Key words: evapotranspiration; land surface model; bias correction; CLM





## 1. Introduction

Land surface models are widely used tools in simulating and predicting the Earth's water and energy budgets over a wide range of spatiotemporal scales (Rodell et al., 2004, Haddeland et al. 2011, Getirana, 2014, Xia et al. 2012a, b, Xia et al. 2016a, b). For example, the Global Land Data Assimilation System (GLDAS) was designed to simulate the terrestrial water and energy budgets over the globe using multiple land surface models (Rodell et al., 2004); and its regional counterpart, the North America Land Data Assimilation System (NLDAS), utilizes four land surface models and focuses on the conterminous United States at a much higher resolution (Rodell et al., 2004, Xia et al. 2012a, b). Products from these two operational systems have been widely used in estimating terrestrial water storage changes (Syed et al. 2008), investigating land-atmosphere coupling strength (Spennemann and Saulo, 2015), analyzing soil moisture variability (Cheng et al. 2015), studying the impact of soil moisture on dust outbreaks (Kim and Choi 2015), and improving data quality of in-situ soil moisture observations (Dorigo et al. 2013, Xia et al. 2015). These model-based estimates of land surface fluxes and state variables are considered important surrogate for observations, as observational data for some components of the global water and energy cycles are scarce in many regions of the world, and are not spatially and temporally continuous where they do exist. However, land surface models are subject to large uncertainties. Haddeland et al. (2011) compared eleven models in simulating evapotranspiration (ET) and found that the range across model is very wide. The global ET on land surface ranges from 415 to 586 mm year$^{-1}$, and the runoff ranges from 290 to 457 mm year$^{-1}$. Xia et al. (2012a-b, 2016a-b) documented large disparity among the four models in NLDAS phase 2 (NLDAS-2) at both the continental and basin scales. The Mosaic and SAC models tend to overestimate ET, whereas the Noah and VIC models are likely to underestimate ET.



Great efforts have been made to improve model performance over the years, through enhancing
both the model parameterization of land surface processes and the model input data. For instance,
during the past ten years, the Community Land Model (CLM) has been upgraded from version 2
to version 4.5 (Bonan et al. 2002, Oleson et al. 2008, Oleson et al. 2013), accompanied by
increasingly accurate and high resolution surface datasets (Lawrence et al. 2011). Comparison with
observations of runoff, evapotranspiration, and total water storage demonstrated continuous
improvement of the model performance (Lawrence et al. 2011). The Noah model is another
example of continuous upgrade from its original version since 1980s (Mahrt et al. 1984). Recent
model developments were on vegetation canopy energy balance, the layered snowpack, frozen soil
and infiltration, soil moisture-groundwater interaction and related runoff production, and
vegetation phenology (Niu et al. 2011). Despite the improved understanding and parameterization
of physical processes and better input data, substantial model biases remain (e.g., Parr et al. 2016,
Wang et al. 2016).
Another approach to reducing model biases is through data assimilation, by merging
observational data and land surface models to obtain optimal estimates for next time step. Fusing
soil moisture observations into land surface model is a typical practice in land data assimilation,
and it has been reported that data assimilation of soil moisture helped in reducing model bias
(Reichle and Koster 2005, Kumar et al. 2008, Yin et al. 2015). However, data assimilation is a
computationally intense task, especially when implementing a multi-model ensemble approach.
Moreover, data assimilation approach is not applicable to future prediction. Parr et al. (2015)
proposed an alternative approach to reducing model biases, and applied it to the Variable
Infiltration Capacity (VIC) model over the Connecticut River Basin for both past simulations and
future projections. The Parr et al. (2015) approach assumes that the relationship between the model



evapotranspiration (ET) and observational ET remain unchanged from one period to another, and
hence the relationship estimated from the calibration period can be used to correct the ET biases
and their effects for any period, historically or in the future. When applied to VIC over the
Connecticut River Basin, Parr et al. (2015) found that the ET bias correction approach significantly
reduces systematic biases in the estimates of both past ET and past river flow, and qualitatively
influences the projected future changes in drought and flood risks.
To establish the robustness of the Parr et al. (2015) method, it needs to be evaluated over
different regions and different climate regimes based on different models. In this study, we
implement the Parr et al. approach into CLM4.5 and evaluate its performance over the whole
CONUS. The land surface model, study area, and the bias correction method are introduced in
Section 2. The data for model calibration and validation, including dataset of ET, runoff, soil
moisture, is described in Section 3. Section 4 presents the calibration and validation results. Finally,
the main findings are summarized and discussed in Section 5.
**2 Model and Methodology**
2.1 Model and Forcing Data
CLM4.5 in its offline mode with the prescribed vegetation phenology is used in this study. The
land surface dataset used in CLM4.5 is derived from different sources. The soil texture data are taken
from Bonan et al. (2012), which was generated using the International Geopshere-Biosphere
Programme soil data (Global Soil Data Task, 2000). Both the percentage of PFTs and the leaf area
index within each grid cell are derived from Moderate Resolution Imaging Spectroradiometer (MODIS)
satellite data (Lawrence et al. 2011). Slope and elevation are obtained from the U.S. Geological Survey
HYDRO1K 1 km data set (Verdin and Greenlee, 1996). Parr et al. (2016) found that CLM4.5 can
realistically capture the spatial pattern of ET over CONUS when the model is forced by the



NLDAS-2 meteorological variables. The spatial correlation coefficients between the simulated
annual ET and the FLUXNET-based observations are as high as 0.93. Wang et al. (2016), using
multiple atmospheric forcing datasets, also reported that CLM4.5 can reasonably reproduce large-
scale pattern of runoff and ET. In this study CLM4.5 is forced by the NLDAS-2 meteorological
forcing (Xia et al., 2012a). NLDAS-2 forcing is available during 1979-present at hourly resolution
on a 0.125° grid system, but is aggregated to a 0.25° resolution in this study as the driving forcing
for CLM4.5. The Conterminous United States (CONUS) is chosen as the study domain over the
globe for the high quality of atmospheric forcing data in this region.
2.3 Methodology

The division of CONUS into Northwest, Southwest, Northeast, and Southeast, which is

based on the 40°N latitude line and the 98°W longitude line, is defined by Lohmann et al. (2004).
This division was later adopted by Xia et al. (2012a) and Tian et al. (2014) when land surface
models were evaluated over CONUS. We follows this division in this study, as shown in Figure 1.

Although land surface models are cable of capturing large-scale pattern of ET, significant

biases were found at finer spatiotemporal scales (Parr et al. 2015, Parr et al. 2016, and Wang et al.
2016), which propagates to influence other components of the hydrological cycle including runoff
and soil moisture (Parr et al. 2015). Following Parr et al. (2015), we derived the climatology of
modeled ET for each model grid cell and for each month based on a simulation during the
calibration period and climatology of observational ET from satellite-based ET data at the same
spatiotemporal resolution during the same period, and estimate the scaling factor between
observational ET and the model ET. This scaling factor, which has its unique spatial variability
and seasonal cycle, is assumed to be time-invariant at the inter-annual and longer time scales. To
correct the ET biases in model simulations during any period, two types of simulations are



conducted sequentially. In the first type of simulation, named as CLM, we run the default CLM4.5
and save the output for three component of ET, i.e., interception loss, plant transpiration, and soil
evaporation, at the PFT level for every time step. The corrected interception loss, plant
transpiration, and soil evaporation are then derived by multiplying the simulated values with the
ET scaling factor, and will be used as input for the second type of simulation, named as CLMET.
In CLMET, we re-run CLM4.5 for the same period as in the first type, but overwrite the three ET
components simulated by the model with the corrected values. Since ET simulations affect the
partitioning of precipitation between ET and runoff, the bias correction in ET is expected to have
direct positive impact on runoff generation and therefore soil moisture.

In this study, we use 1986-1995 as the calibration period and 2000-2014 as the validation

period. The simulations during the calibration period are obtained from a 16-year (1980-1995)
CLM run with the first 6-year run disregarded as the spinup. Both CLM and CLMET runs during
the validation period starts with the initial condition of January $1^{st}$ 1996 obtained from the
calibration period. Since the overwriting process in CLMET may break the water balance, the
model checks if the interception loss exceeds the water stored in vegetation canopy and if the
surface soil water is sufficient to support soil evaporation, and makes adjustment if needed. This
minimizes the unbalance caused by overwriting ET components in CLMET.
**3 Data**
3.1 ET
3.1.1 GLEAM ET

GLEAM (The Global Land Evaporation Amsterdam Model) version 3.0a (Miralles et al.

2011, Martens et al. 2016) is used to calibrate the ET scaling factors and to validate CLM and
CLMET. GLEAM 3.0 has three subsets, i.e., 3.0a, 3.0b, and 3.0c. GLEAM 3.0a is derived based

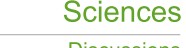
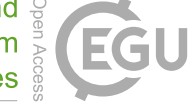

on reanalysis net radiation and air temperature, a combination of gauge-based, reanalysis and
satellite-based precipitation and satellite-based vegetation optical depth, spanning the 35-year
period 1980–2014 (http://www.gleam.eu/). Potential evaporation in GLEAM 3.0 is calculated
using a Priestley and Taylor equation based surface net radiation and near-surface air temperature,
and is converted to actual evaporation based on the multiplicative evaporative stress factor. The
dataset has been used in studying soil moisture-temperature coupling (Miralles et al. 2012), the
impact of land surface on precipitation (Guillod et al. 2015), and the climate control on land surface
evaporation (Miralles et al., 2014). Recent evaluations conducted at both tower and global scales
shows that GLEAM-based ET is superior to MODIS-based and the Surface Energy Balance
System (SEBS) based ET products (Michel et al. 2016, Miralles et al. 2016). The spatial resolution
for GLEAM dataset is 0.25°, which is consistent with the resolution of CLM4.5 used in this study.
The temporal resolution of GELAM dataset is daily, and the monthly aggregated ET is used to
derive the scaling factors.
3.1.2 MODIS and FLUXNET-MTE ET

Another two ET products are used for independent evaluations: MODIS ET and

FLUXNET-MTE (model tree ensemble) ET. Mu et al. (2007, 2011) produced a MODIS-based
global ET dataset by revising the Penman–Monteith (PM) equation. The dataset is arguably the
most widely used remote-sensing-based global ET product (Miralles et al. 2016). Monthly version
of the MODIS-based product at the 0.5° spatial resolution are used to validate the model with the
bias correction method. The FLUXNET-MTE global ET dataset was derived from 253 FLUXNET
eddy covariance towers distributed over the globe using the model tree ensemble (MTE) approach
(Jung et al., 2009, 2010). The record gaps of half hourly eddy covariance fluxes were filled first,
and the complete tower-based dataset is used to train MTE to produce monthly global ET dataset





at 0.5° spatial resolution. The data have been used to study the ET trend (Jung et al., 2010) and to
improve canopy processes in a land surface model (Bonan et al., 2011). As FLUXNET sites over
the CONUS are very dense, the quality of the FLUXNET-MTE dataset in our study domain is
expected to be high. The MODIS dataset is available from 2000-2014, and the FLUXNET-MTE
dataset is available from 1982-2011. We chose the overlap period of those two products, 2000-
2011, for model validations using MODIS and FLUXNET-MTE dataset.
3.2 Observation-based Runoff Coefficient
The runoff coefficient (the ratio of runoff to precipitation) of Global Streamflow
Characteristics Dataset (GSCD) version 1.9 (Beck et al., 2013, Beck et al., 2015) is used to evaluate
the model performance in simulating runoff. The GSCD dataset was produced based on
streamflow observations from approximately 7500 catchments over the globe. A data-driven
approach was adopted to derive the gridded streamflow characteristics at the 0.125° resolution on
a global scale. This dataset is relatively reliable for the grid cells within which a large number of
catchments data is used. The uncertainty is low in North America, Europe, and southeastern
Australia where a large number of observations are available.
3.3 In-situ soil moisture observations
The North American Soil Moisture Database (NASMD) is used to evaluate the model
performance in simulating soil moisture in both the surface (0-10cm) and root-zone (0-100cm)
layers. NASMD was initiated in 2011 to provide support for developing climate forecasting tools,
calibrating land surface models, and validating satellite-derived soil moisture algorithms. A
homogenized procedure has been implemented, as the measurement stations are across a variety
of in-situ networks. In addition, a quality control (QC) algorithm was applied to the measurement
records. Liao et al. (submitted to Journal of Hydrometeorology, 2016) developed an additional QC





algorithm to further improve data quality of the NASMD soil moisture based on the approach of
Xia et al. (2015). The soil moisture after QC agree more closely with a manual-checked benchmark.
More details on the QC algorithm and the comparison with the benchmark can be found in Liao et
al. (2016). The in-situ observations in the states of Alabama (AL), Illinois (IL), Mississippi (MS),
Nebraska (NE), and Oklahoma (OK) from 2006-2010 are selected from NASMD, as a large
number stations is evenly distributed over these states and observation records during this period
are relatively complete after QC. The numbers of stations in AL, IL, MS, NE, and OK are 10, 19,
14, 45, 105, and 39, respectively, as shown in Figure 2. Since the soil layer in which measurement
is conducted varies with stations, we interpolate the volumetric soil water content to the 5 cm and
50 cm depth for all stations using the liner interpolation method to compare with the modeled soil
moisture in the 0-10 cm and 0-100 cm layers.
**4 Results**
4.1 ET scaling factor calibration

Figure 3 shows the climatological scaling factors for each month over CONUS based on

the 1986-1995 period. The model simulations generally agree better with observations during the
warm seasons, whereas the difference between simulations and observations are large during the
cold seasons. The scaling factors greatly vary with region, as indicated by area-averaged values
for four sub regions. For instance, the area-averaged values are 0.41, 0.58, 0.29, and 0.52 for
Northwest, Southwest, Northeast, and Southeast in November, respectively. The overestimation is
overwhelming during October, November, December, and January, whereas underestimation
occurs in many areas during March, April, and May. The overestimation is very severe with
simulations being almost 5 times of observations for Northeast CONUS in December.
4.2 Evaluation





We evaluate the effectiveness of the ET bias correction approach in CLM4.5 by comparing
results from CLM and CLMET with observations. The evaluation metrics examined include bias,
relative bias, root mean square error (RMSE), and correlation coefficient (R). Since the spatial
resolution of some observational data is not consistent with the model resolution, we upscale the
finer resolution data to match the coarser resolution data using simple arithmetic averages.  For
example, when the MODIS and FLUXNET-MTE ET are used for validation, we aggregate the
four 0.25° modeled ET within each 0.5° grid cell; for the GSCD runoff data, we aggregate
observations from 0.125° to 0.25° to match the model resolution. As in-situ soil moisture
observations are essentially on the point scale, we spatially average observed soil moisture in each
state and compare the averaged observations with the averaged model simulations over grid cells
within the same state.
4.2.1 ET
Figure 4 shows the multi-year averages (2000-2014) of ET derived from GLEAM,
simulated by CLM and CLMET, and the relative bias of simulations against GLEAM. Since
GLEAM observations are not available in many areas in December and January (Figure 3), these
areas are left blank in Figure 4. Over most of CONUS, CLM overestimates ET relative to GLEAM
data, and CLMET reduces ET as well as ET biases. The averaged relative bias in CLM over
CONUS is 9.06%, with relative bias exceeding 10% in a substantial portion of CONUS; and in
CLMET, the CONUS-averaged relative bias is reduced to -2.05%, and it is within 10% over most
of CONUS. This improvement is more significant over eastern CONUS than western CONUS.
Table 1 shows the statistics on the model performance with these two schemes during different
seasons and in four sub regions. CLM overestimates the CONUS-averaged ET in all other seasons
except for March-April-May (MAM), and the largest overestimation occurs in Southeast CONUS





during December-January-February (DJF) with a relative bias as large as 135.1%. The
underestimation in MAM is largest over Southwest CONUS with a relative bias of -17.9%.
CLMET substantially improves the model performance as indicated by the various metrics. All
the statistics in CLMET is superior to those in CLM with a few exceptions in bias or relative bias.
The improvement from CLM to CLMET is more substantial for September-October-November
(SON) and DJF than MAM and June-July-August (JJA). The relative bias of 43.4% (54%) in CLM
is reduced to 5% (7.8%) in CLMET over CONUS during SON (DJF). For the regional comparison,
the improvement is greatest over Southeast CONUS. All the positive biases in all seasons over
Southeast CONUS are significantly reduced.

To understand the differences between CLM and CLMET, we select four months from

each of seasons, January, April, July, and November, to examine the relationship between the
relative bias of model simulations and the scaling factor changes from calibration period (1986-
1995) to validation period (2000-2014) in Figure 5. The improvement from CLM to CLMET is
evident, especially in January and November (Figure 5a-b). Although the bias is dramatically
reduced in CLMET, it remains large in Northeast CONUS in January (Figure 5b1). In addition,
the bias in CLMET seems larger in western CONUS than eastern CONUS (Figure 5b). The spatial
patterns of the relative biases in CLMET and the scaling factor differences between the two periods
demonstrate a great degree of similarity (Figure 5b-5c), and the scatter plots between the two
quantities (Figure 5d) reflect a strong correlation. This suggests that the degree to which CLMET
can improve model performance in simulating ET greatly depends on how stable the scaling factor
is from the calibration period to the validation period, i.e., how well the assumption of a time-
invariant scaling relationship holds. Over most of CONUS, changes in the scaling factor are within

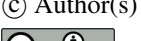



10% (Figure 5d). This temporal stability of the relationship between observed ET and simulations
guarantees improvements from CLM to CLMET.

The analysis on time series of ET from GLEAM and two types of simulations also

demonstrates improvement from CLM to CLMET. Climatological seasonal cycles of ET over
CONUS and four sub regions for 2000-2014 are shown in Figure 6. The improvement from CLM
to CLMET is more evident in SON and DJF, which is consistent with the spatial analysis. The
simulated ET from CLMET is very close to GLEAM observations over most seasons. However,
underestimate of ET in CLMET in western CONUS during summer still exists. For example,
simulation is lower than observation in Northwest CONUS in July (Figure 6b), and Southwest
CONUS in May (Figure 6c). Figure 7 shows the temporal evolution of the simulated ET in CLM
and CLMET against GELAM observations over COUNS and four sub regions. It is evident that
the bias correction method in CLMET is very effective in adjusting overestimation (positive bias).
However, underestimation (negative bias) existing in CLM is sometimes not well corrected. The
difference has to do with how water limits the ET occurrence.  When a lower ET value replace the
positive biased one, the water on land is sufficient to support the reduced ET. By contrast, when a
higher ET value replace the negative biased one, the land surface model checks if the water in soil
layer and vegetation canopy can sustain the elevated ET. The extent to which ET increases is
limited by the availability of water stored in soil layer and vegetation canopy. Therefore, actual
ET after the water availability check in CLMET does not increase much if the water is limited
even through the corrected ET fed into model is larger.

The model performance metrics are also calculated for ET simulation at shorter time scales

(weekly and daily, Figures now shown).  Table 2 summarize RMSE and correlation coefficient of
CLM and CLMET against the GLEAM observations from seasonally to daily. Since the





correlation coefficient (R) is already high in CLM, the improvement from CLM to CLMET
according to R is limited. By contrast, RMSE is greatly changed from CLM to CLMET. The largest
change is found in Southeast CONUS, which is consistent with the model performance in
simulating the spatial pattern of ET. The model performance becomes worse with shorter temporal
scales (from monthly to weekly to daily), as shown in Table 2, which is consistent with findings
of Parr et al. (2015) who also found downgraded model performance with the higher temporal
resolution when the same method is applied to the VIC model in the Connecticut river basin.
In addition, CLM and CLMET performances are also evaluated using two independent
observation dataset of ET, MODIS-based and FLUXNET-based ET (Figure 8, Tables 3 and 4).
For the multi-year averaged ET, the relative bias in CLMET is smaller than that in CLM, and the
improvement is greater in eastern CONUS than western CONUS as compared with either MODIS-
or FLUXNET-based ET. Note that there is still a substantial overestimation in western CONUS in
CLMET compared with the MODIS ET, partially because the algorithm developed by Mu et al.
(2007, 2011) underestimate ET in the MODIS product (Michel et al. 2016, Miralles et al. 2016).
If the reference is the MODIS-based ET, CLMET corrects bias for all other three seasons except
for MAM (Table 3). Bias, relative bias and RMSE in CLMET is greater than CLM for the whole
CNOUS, Northwest, Southwest, and Northeast in MAM. Among all other three seasons, SON is
the reason when model performance is improved most from CLM to CLMET. If the FLUXNET
ET is taken as a reference, the improvement is found in all four sub regions. The improvement in
MAM is minor, whereas the improvement in SON is substantial. The performance in CLMET
against MODIS or FLUXNET is similar to the model performance against GLEAM but with
smaller magnitudes.
4.2.2 Runoff



Using the runoff coefficient (the ratio of runoff to total precipitation) derived from GRDC
as the benchmark, we evaluate the model performance in CLM and CLMET in simulating runoff
(Figure 9). The CONUS averaged runoff coefficient in CLM and CLMET are 0.18 and 0.21, which
is comparable with the GRDC-based runoff coefficient (0.22). However, CLM underestimate the
runoff in most areas of CONUS due to overestimate of ET.  CLMET alleviates the underestimation
by decreasing ET therefore increasing the runoff, especially over eastern CONUS. The relative
bias of CLMET against GRDS is 0.72%, which is much smaller than the value in CLM (-9.21%).
Table 5 shows the regional difference in runoff simulations in CLM and CLMET. The
improvement is greater over Eastern CONUS than Western CONUS, which is consistent with the
improvement of ET simulations. The most striking improvement occurs in Southeast CONUS,
with the relative bias (RMSE) decreased from -24.7% (0.091) to -8.2% (0.06). Because only the
multi-year mean annual runoff coefficient is available for GRDC, we cannot examine the seasonal
dependency of the model performance improvement.
The increase in runoff from CLM to CLMET is mainly due to the increase in subsurface
runoff (Figure 10). The same value of the ET scaling factor within each gird cell are applied to
three components of ET (interception loss, plant transpiration and soil evaporation) in this study.
Because interception loss accounts for a small portion of total ET, the absolute change of
interception loss (decrease from CLM to CLMET over most areas) is much smaller compared with
plant transpiration and soil evaporation (not shown). As a result, the increase in throughfall does
not change much from CLM to CLMET, which leads to smaller increases in surface runoff. By
contrast, plant transpiration and soil evaporation is more significantly reduced by CLMET,
inducing wetter soil and therefore more subsurface runoff.
4.2.3 Soil moisture

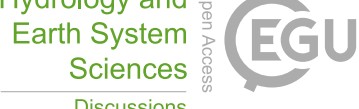



As analyzed in Section 4.2.2, reduction in all three components of ET interception loss,
plant transpiration, and soil evaporation from CLM to CLMET slows down moisture depletion
from the soil.  As a result, the water content at different soil layers increases with the reduced ET.
Figure 11 shows soil water at the surface and root-zone layers simulated from CLM and CLMET,
and their differences during the summer season (JJA). From CLM to CLMET, the changes over
CONUS show an overwhelmingly increase signal for both surface and root-zone soil moisture.
The moisture increase in the top 0-100 cm soil layer from CLM to CLMET in central CONUS is
very evident, which may have significant implications in drought monitoring and assessment. For
example, Central Great Plains experienced a severe drought in summer of 2012, and soil moisture
derived from land surface models was used to evaluate the intensity of the drought event (Hoerling
et al. 2014, Livneh and Hoerling 2016). Unfortunately, land surface models tend to systematically
overestimate drought (Milly and Dunne 2016, Ukkol et al. 2016). The more accurate simulations
of ET and soil moisture resulting from the bias correction method used in this study may prove
useful in better drought monitoring and assessment.
Figures 12 and 13 show the comparisons between observed soil moisture and modeled soil
moisture from CLM and CLMET on the monthly scale during 2006-2010 for the top 0-10 cm and
top 0-100 cm soil, respectively. The soil water increase from CLM to CLMET is more evident
during SON and DJF, which is consistent with changes in ET that also features more decreases
during SON and DJF. Because the soil in CLM shows dry bias over most states with the exception
of soil moisture at the top 10 cm layer in Alabama, CLMET generally alleviate the dry bias in
CLM. Therefore, the RMSE values against the NASMD observations in CLMET is smaller or at
least the same to RMSE values in CLM. An exception exists for the top 0-10 cm layer in Alabama
where a wet bias is found in CLM. The soil water content difference between CLM and CLMET



is larger for the 0-100 cm layer than the 0-10 cm layer, because plant transpiration, to which a
large fraction of ET and therefore a large fraction of ET bias correction are associated, primarily
depletes moisture from the rooting zone which is deeper than 10 cm. As such, the improvement is
more evident for the top 0-100 cm layer. For example, in Mississippi, the RMSE is reduced from
0.048 $m^3$ $m^{-3}$ in CLM to 0.042 in CLMET at the top 0-10 cm layer, and from 0.07 to 0.06 $m^3$ $m^{-3}$
at the top 0-100 cm layer.

**5 Summary and discussions**

In this study, we implemented the on-line bias correction approach proposed by Parr et al.

(2015) to CLM4.5, and evaluated the effectiveness of the approach in reducing model biases over
CONUS. The bias correction algorithm was calibrated using the GLEAM ET product combined
with the default CLM4.5 output over the period of 1986-1995, and was validated over the period
of 2000-2014 using three ET datasets, the GRDC runoff product, and the NASMD soil moisture
data. Results from all evaluation metrics indicate substantial improvement in the estimation of the
terrestrial hydrological cycle.

The degree to which the Parr et al. (2015) approach improves the quantification of the

hydrological cycle differs among the CONUS sub-regions, and is highly related to whether the
fundamental assumption of Parr et al. (2015) (on a time-invariant relationship characterizing the
default model biases) holds or not. Although the scaling factors between observations and
simulations do not change much from the calibration period to the validation period over most
regions in most seasons, dramatic changes do exist in some areas. Differences in the scaling factors
between the calibration and verification/application periods greatly influence the effectiveness of
the bias correction method, with large differences causing the approach to be less effective leaving
substantial biases in CLMET. Northeast CONUS during winter is an example of having a large





bias in CLMET due to greater changes in the ET scaling factor from the calibration period to the
verification period. Overall, the approach reduces land surface dry biases over eastern CONUS in
CLM4.5.

For a given grid cell and given month, the scaling factors for all three ET components, i.e.,

interception loss, plan transpiration, soil evaporation, are the same in this study, set to be the ratio
of the remote sensing ET to the modeled ET. Since the GLEAM dataset contains values of three
components besides total ET, we conducted additional experiments in which the scaling factors
for each ET component was estimated separately, using the ratio of each ET component from the
GLEAM product to the corresponding ET component from CLM during the same calibration
period.   However, results based on the component-specific scaling factors do not show any
improvement, which is likely caused by the inaccurate partitioning of ET into interception loss,
plan transpiration, soil evaporation. Miralles et al. (2016) compared the ET partitioning for three
widely used remote sensing based ET products, and found that the contribution of each component
to ET is dramatically different among these three products. For instance, the percentage of global
ET accounted for by soil evaporation ranges from 14% to 52%, and the ranges are even larger at
the regional and local scales. Because the in-situ measurements of separate components of ET is
very scarce, it is particularly challenging to validate the accuracy of the remote sensing based
estimated of the three ET components. These challenges led Miralles et al. (2016) to recommend
against the use of any single product in partitioning ET.

The bias correction method evaluated in this study can effectively improves the estimates

of surface fluxes and state variables in the absence of improved physical parameterizations in land
surface models. It is applicable to not only historical simulations but also future predictions (Parr
et al. 2015). It provides an alternative approach to, but would in no way replace, model





improvement through better parameterization of physical processes. Development of better
physical parameterizations has to be based on improved understanding of physical processes, more
effective mathematical formulations, and higher quality surface type dataset, which requires a
long-term commitment from the land surface modeling community.

**6. Data availability**
The GLEAM ET data was provided by the GLEAM team at the website www.GLEAM.eu. The
MODIS ET data by NTSG, University of Montana at the website
http://www.ntsg.umt.edu/project/mod16. The FLUXNET-MTE ET data was provided by Max
Planck Institute for Biogeochemistry at the website https://www.bgc-
jena.mpg.de/geodb/projects/Data.php. The GSCD runoff data was provided by the Amsterdam
Critical Zone Hydrology Group at the website http://hydrology-
amsterdam.nl/valorisation/GSCD.html. The original NASMD soil moisture data is available at the
website http://soilmoisture.tamu.edu/. The quality-controlled NASMD soil moisture data can be
obtained from the authors upon request.

**Author contributions**
D. Wang and G. Wang designed the study. D. Wang conducted model simulations and data
analysis with input from G. Wang, D. Parr and C. Fu, D. Wang and G. Wang wrote the paper with
input from Y. Xia. W. Liao and Y. Xia contributed to data processing.

**Competing interests**
The authors declare that they have no conflict of interest.






## Acknowledgements

This study is supported by National Natural Science Foundation of China (Grant No. 51379224),
and the Fundamental Research Funds for the Central Universities.

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





Table 1 Spatial evaluations of simulated ET from two different types of runs (CLM and
CLMET) against GLEAM observations over COUNS, Northwest (NW), Southwest (SW),
Northeast (NW), and Southeast (SW) annually and seasonally for 2000-2014. March-April-May:
MAM, June-July-August: JJA, September-October-November: SON, December-January-
February: DJF

| Season | Region | Bias (mm day$^{-1}$) | | Relative bias (%) | | RMSE (mm day$^{-1}$) | |
|---|---|---|---|---|---|---|---|
| | | CLM | CLMET | CLM | CLMET | CLM | CLMET |
| Annual | CONUS | 0.141 | -0.027 | 9.2 | -2.0 | 0.301 | 0.157 |
| | NW | -0.227 | -0.245 | -11.1 | -13.7 | 0.382 | 0.329 |
| | SW | 0.065 | -0.035 | 9.2 | -3.6 | 0.185 | 0.121 |
| | NE | 0.117 | -0.017 | 8.0 | -0.4 | 0.255 | 0.138 |
| | SE | 0.315 | 0.041 | 15.6 | 2.1 | 0.355 | 0.099 |
| MAM | CONUS | -0.081 | -0.062 | -5.8 | -3.3 | 0.351 | 0.228 |
| | NW | -0.138 | -0.074 | -6.7 | -2.7 | 0.326 | 0.244 |
| | SW | -0.211 | -0.122 | -17.9 | -9.4 | 0.318 | 0.206 |
| | NE | -0.191 | -0.079 | -8.3 | -2.8 | 0.429 | 0.293 |
| | SE | 0.19 | 0.022 | 8.9 | 1.5 | 0.346 | 0.165 |
| JJA | CONUS | 0.094 | -0.041 | 6.4 | -1.4 | 0.451 | 0.332 |
| | NW | -0.137 | -0.121 | -3.9 | -4.0 | 0.487 | 0.408 |
| | SW | 0.147 | -0.006 | 18.3 | -0.9 | 0.352 | 0.232 |
| | NE | 0.045 | -0.124 | 2.5 | -2.7 | 0.55 | 0.452 |
| | SE | 0.332 | 0.075 | 9.1 | 2.1 | 0.414 | 0.181 |
| SON | CONUS | 0.361 | 0.049 | 43.4 | 5.0 | 0.434 | 0.159 |
| | NW | 0.216 | 0.005 | 55.9 | 3.4 | 0.328 | 0.154 |
| | SW | 0.23 | 0.045 | 39.5 | 5.2 | 0.283 | 0.118 |
| | NE | 0.482 | 0.079 | 49.5 | 7.3 | 0.53 | 0.247 |
| | SE | 0.499 | 0.061 | 34.5 | 4.1 | 0.531 | 0.11 |
| DJF | CONUS | 0.183 | -0.002 | 54.0 | 7.8 | 0.278 | 0.121 |
| | NW | 0.039 | -0.088 | 32.9 | -8.3 | 0.305 | 0.165 |
| | SW | 0.132 | -0.013 | 35.7 | -1.3 | 0.192 | 0.069 |
| | NE | 0.267 | 0.09 | 135.1 | 61.3 | 0.374 | 0.24 |
| | SE | 0.24 | 0.004 | 49.2 | 2.8 | 0.292 | 0.072 |







Table 2. Temporal evaluations of simulated ET against GLEAM observations over COUNS,
Northwest (NW), Southwest (SW), Northeast (NW), and Southeast (SE) at different temporal
scales for the period of 2000-2014.

| Region | Season | RMSE (mm day⁻¹) | | Correlation coefficient | |
|---|---|---|---|---|---|
| | | CLM | CLMET | CLM | CLMET |
| CONUS | Climatologically seasonal | 0.224 | 0.049 | 0.983 | 0.999 |
| | Monthly | 0.231 | 0.078 | 0.981 | 0.997 |
| | Weekly | 0.252 | 0.125 | 0.976 | 0.992 |
| | Daily | 0.281 | 0.172 | 0.967 | 0.984 |
| NW | Climatologically seasonal | 0.183 | 0.095 | 0.981 | 0.996 |
| | Monthly | 0.209 | 0.128 | 0.973 | 0.989 |
| | Weekly | 0.251 | 0.251 | 0.96 | 0.975 |
| | Daily | 0.307 | 0.256 | 0.936 | 0.954 |
| SW | Climatologically seasonal | 0.197 | 0.077 | 0.92 | 0.988 |
| | Monthly | 0.218 | 0.113 | 0.91 | 0.974 |
| | Weekly | 0.252 | 0.161 | 0.887 | 0.952 |
| | Daily | 0.298 | 0.222 | 0.853 | 0.916 |
| NE | Climatologically seasonal | 0.314 | 0.1 | 0.98 | 0.999 |
| | Monthly | 0.325 | 0.152 | 0.977 | 0.995 |
| | Weekly | 0.381 | 0.245 | 0.967 | 0.986 |
| | Daily | 0.467 | 0.363 | 0.947 | 0.966 |
| SE | Climatologically seasonal | 0.347 | 0.061 | 0.993 | 1 |
| | Monthly | 0.374 | 0.139 | 0.987 | 0.995 |
| | Weekly | 0.414 | 0.209 | 0.978 | 0.988 |
| | Daily | 0.493 | 0.325 | 0.958 | 0.97 |










Table 3. Same as the Table 1 except simulation against with MODIS observations and for the

period of 2000-2011.

| Season | Region | Bias (mm day$^{-1}$) | | Relative bias (%) | | RMSE (mm day$^{-1}$) | |
|---|---|---|---|---|---|---|---|
| | | CLM | CLMET | CLM | CLMET | CLM | CLMET |
| Annual | CONUS | 0.321 | 0.184 | 30.8 | 19.9 | 0.427 | 0.325 |
| | NW | 0.28 | 0.234 | 35.8 | 29.5 | 0.367 | 0.334 |
| | SW | 0.282 | 0.188 | 39.7 | 26.4 | 0.428 | 0.364 |
| | NE | 0.278 | 0.136 | 19.6 | 9.8 | 0.316 | 0.199 |
| | SE | 0.431 | 0.16 | 24.9 | 10.6 | 0.538 | 0.348 |
| MAM | CONUS | 0.514 | 0.533 | 50.1 | 55.8 | 0.631 | 0.635 |
| | NW | 0.564 | 0.628 | 67.2 | 74.4 | 0.636 | 0.687 |
| | SW | 0.345 | 0.438 | 45.9 | 61.8 | 0.538 | 0.599 |
| | NE | 0.547 | 0.654 | 51.7 | 61.8 | 0.58 | 0.675 |
| | SE | 0.596 | 0.436 | 34.6 | 25.8 | 0.735 | 0.578 |
| JJA | CONUS | 0.251 | 0.115 | 18.2 | 12.1 | 0.759 | 0.691 |
| | NW | 0.263 | 0.281 | 23.8 | 25.5 | 0.704 | 0.71 |
| | SW | 0.344 | 0.192 | 28.8 | 14.4 | 0.806 | 0.724 |
| | NE | 0.028 | -0.145 | 2.9 | -2.4 | 0.662 | 0.564 |
| | SE | 0.31 | 0.052 | 13.2 | 5.8 | 0.829 | 0.72 |
| SON | CONUS | 0.345 | 0.045 | 48.2 | 11.0 | 0.459 | 0.285 |
| | NW | 0.261 | 0.056 | 56.8 | 13.2 | 0.369 | 0.263 |
| | SW | 0.284 | 0.096 | 55.9 | 20.9 | 0.43 | 0.306 |
| | NE | 0.448 | 0.048 | 47.4 | 6.2 | 0.483 | 0.209 |
| | SE | 0.417 | -0.019 | 32.1 | 2.7 | 0.547 | 0.329 |
| DJF | CONUS | 0.173 | 0.041 | 85.2 | 41.6 | 0.384 | 0.278 |
| | NW | 0.027 | -0.031 | 88.7 | 65.5 | 0.385 | 0.362 |
| | SW | 0.156 | 0.028 | 70.5 | 25.4 | 0.292 | 0.18 |
| | NE | 0.091 | -0.014 | 96.4 | 38.5 | 0.344 | 0.236 |
| | SE | 0.403 | 0.17 | 87.5 | 33.9 | 0.474 | 0.281 |











Table 4. Same as the Table 3 except simulation against with FLUXNET observations.

| Season | Region | Bias (mm day$^{-1}$) | | Relative bias (%) | | RMSE (mm day$^{-1}$) | |
|---|---|---|---|---|---|---|---|
| | | CLM | CLMET | CLM | CLMET | CLM | CLMET |
| Annual | CONUS | 0.207 | 0.072 | 13.3 | 3.9 | 0.328 | 0.242 |
| | NW | 0.07 | 0.025 | 5.8 | 1.2 | 0.222 | 0.233 |
| | SW | 0.051 | -0.042 | 6.8 | -4.1 | 0.244 | 0.241 |
| | NE | 0.309 | 0.175 | 21.9 | 13.0 | 0.334 | 0.248 |
| | SE | 0.427 | 0.154 | 21.3 | 7.6 | 0.461 | 0.248 |
| MAM | CONUS | 0.27 | 0.291 | 15.8 | 19.5 | 0.418 | 0.399 |
| | NW | 0.266 | 0.33 | 22.4 | 28.0 | 0.349 | 0.401 |
| | SW | -0.042 | 0.051 | -7.3 | 2.5 | 0.298 | 0.301 |
| | NE | 0.288 | 0.401 | 21.6 | 30.4 | 0.338 | 0.434 |
| | SE | 0.561 | 0.399 | 26.4 | 18.5 | 0.6 | 0.448 |
| JJA | CONUS | 0.197 | 0.063 | 7.0 | 0.5 | 0.608 | 0.517 |
| | NW | -0.149 | -0.131 | -8.7 | -7.6 | 0.506 | 0.506 |
| | SW | 0.029 | -0.122 | 9.2 | -6.1 | 0.594 | 0.555 |
| | NE | 0.415 | 0.257 | 13.6 | 8.8 | 0.492 | 0.369 |
| | SE | 0.565 | 0.304 | 16.9 | 9.4 | 0.779 | 0.585 |
| SON | CONUS | 0.216 | -0.081 | 20.3 | -8.5 | 0.353 | 0.291 |
| | NW | 0.072 | -0.132 | 9.2 | -20.0 | 0.224 | 0.275 |
| | SW | 0.132 | -0.055 | 21.1 | -5.2 | 0.311 | 0.277 |
| | NE | 0.356 | -0.03 | 33.7 | -0.6 | 0.473 | 0.386 |
| | SE | 0.346 | -0.091 | 21.2 | -5.4 | 0.396 | 0.23 |
| DJF | CONUS | 0.144 | 0.014 | 38.0 | 5.4 | 0.266 | 0.189 |
| | NW | 0.09 | 0.033 | 20.6 | 0.5 | 0.271 | 0.247 |
| | SW | 0.086 | -0.042 | 20.9 | -8.0 | 0.17 | 0.12 |
| | NE | 0.175 | 0.073 | 78.3 | 35.6 | 0.329 | 0.228 |
| | SE | 0.236 | 0.003 | 42.8 | 1.0 | 0.282 | 0.128 |











Table 5 Statistics of simulated annual runoff coefficient (ratio of runoff to total precipitation)
against GRDC observations over COUNS, and Northwest (NW), Southwest (SW), Northeast
(NW), and Southeast (SW) for the period of 2000-2014.

| | Bias | | Relative bias (%) | | RMSE | |
|---|---|---|---|---|---|---|
| | CLM | CLMET | CLM | CLMET | CLM | CLMET |
| CONUS | -0.053 | -0.028 | -18.5 | -7.3 | 0.198 | 0.192 |
| Northwest | -0.046 | -0.038 | -13.5 | -7.0 | 0.146 | 0.145 |
| Southwest | -0.026 | -0.02 | -19.9 | -11.8 | 0.373 | 0.373 |
| Northeast | -0.06 | -0.023 | -15.7 | -2.1 | 0.108 | 0.094 |
| Southeast | -0.074 | -0.026 | -24.7 | -8.2 | 0.091 | 0.06 |















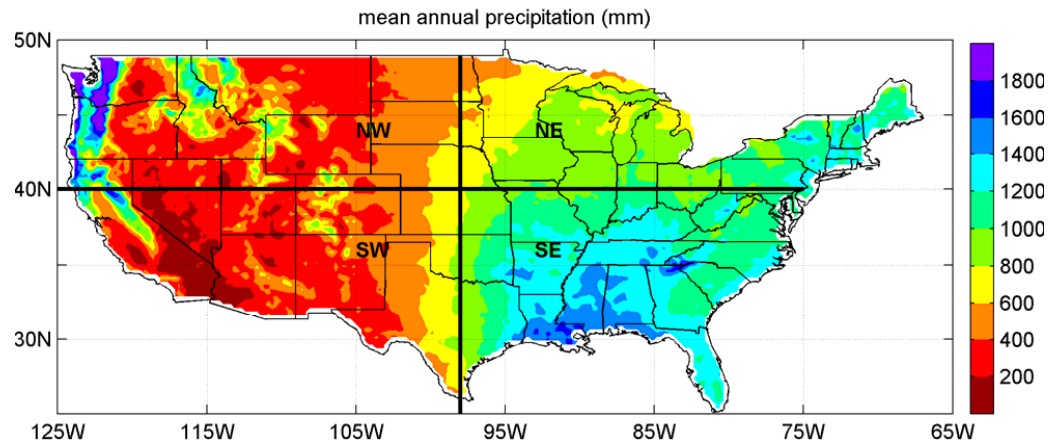


660  Figure 1 Mean annual (1980-2015) precipitation in mm over conterminous USA (CONUS).

661  NW, SW, NE, and SE represent Northwest, Southwest, Northeast, and Southeast, respectively.

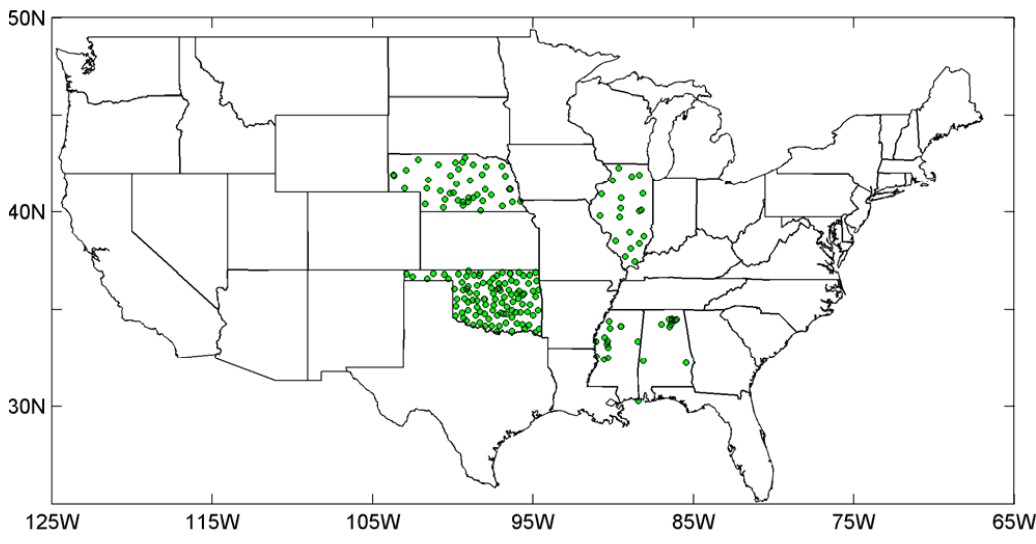


663  Figure 2 Location of in-situ soil moisture observations in the states of Alabama, Illinois,

664       Mississippi, Nebraska, and Oklahoma.






Figure 3 Scaling factors of the CLM simulated ET to the GLEAM ET for each month during
1986-1995. The numbers in titles are CONUS-averaged values, and the number of within figures
are area-averaged values for each of four sub regions (NW, SW, NE, and SE).





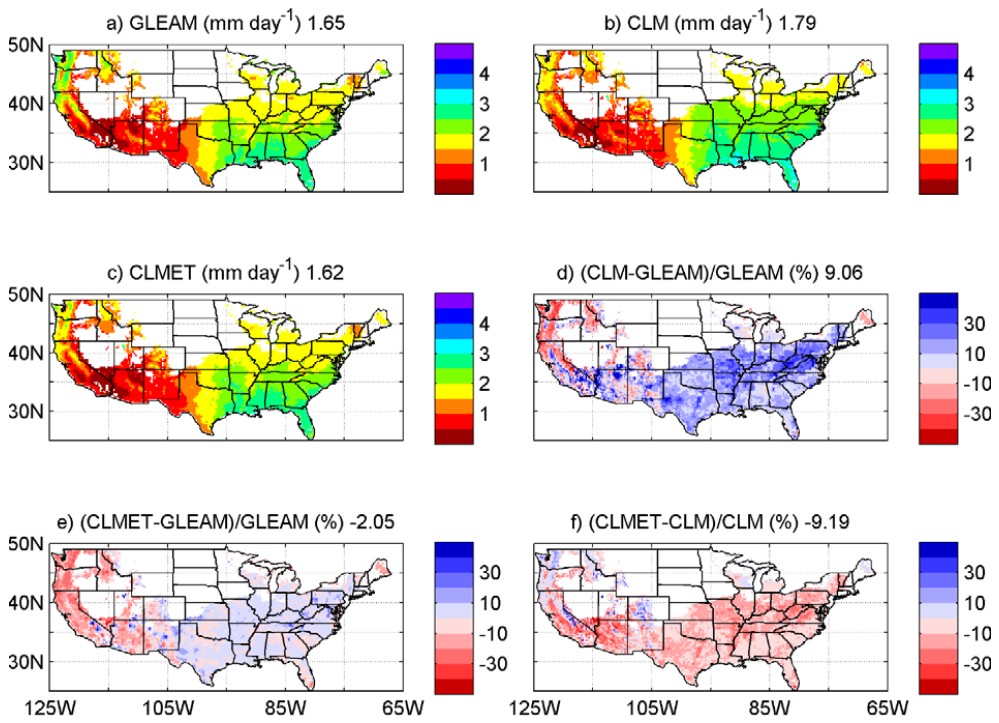

Figure 4 Mean annual ET from a) GLEAM, b) CLM, and c) CLMET, and the relative

differences between d) CLM and GLEAM, e) CLMET and GLEAM, and f) CLMET and CLM

during 2000-2014. Numbers in titles are CONUS-averaged values.





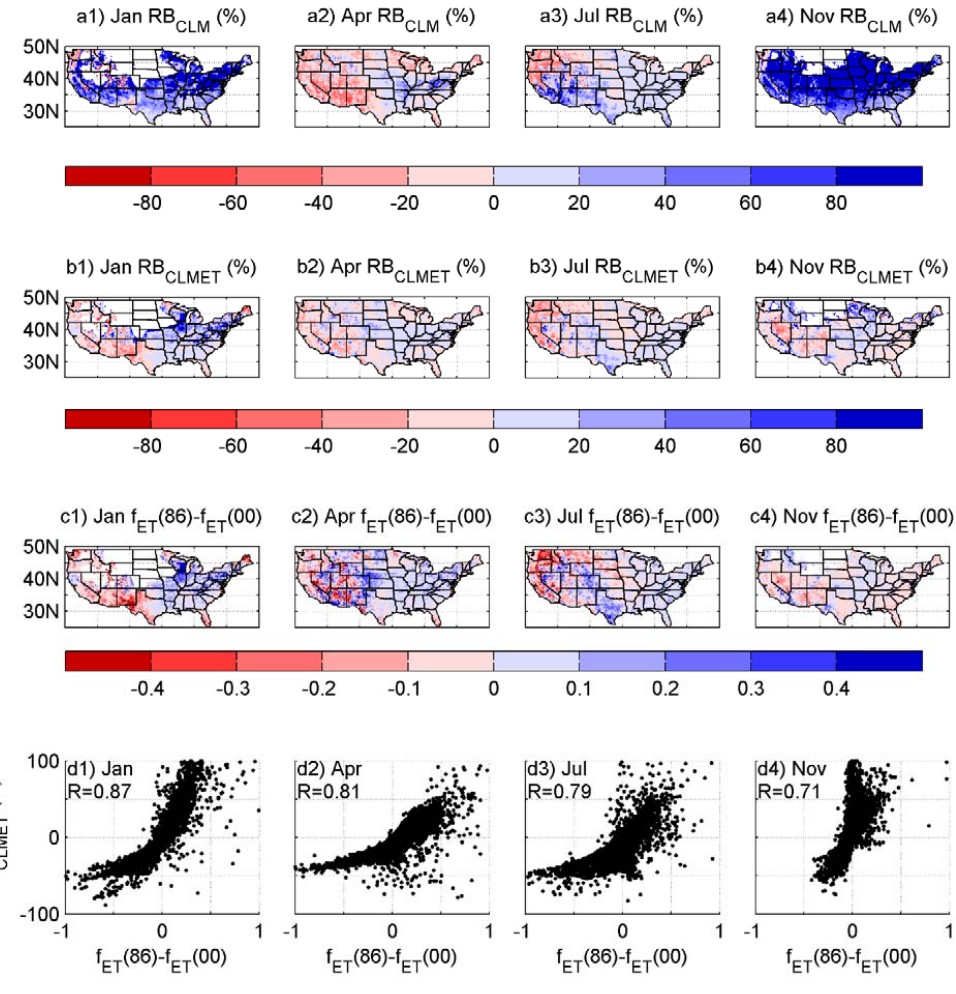


Figure 5 Relative bias (RB) for CLM (RB$_{CLM}$), RB for CLMET (RB$_{CLMET}$), difference in scaling
factor f$_{ET}$ between the period 1986-1995 and the period 2000-2014 (f$_{ET}$(86)- f$_{ET}$(00)), and scatter
plots of f$_{ET}$(86)- f$_{ET}$(00) versus RB$_{CLMET}$ in January (Jan), April (Apr), July (Jul), and November

(Nov).








Figure 6 Seasonal cycles of ET from GLEAM, CLM, and CLMET over CONUS, Northwest,

Southwest, Northeast, and Southeast during 2000-2014.





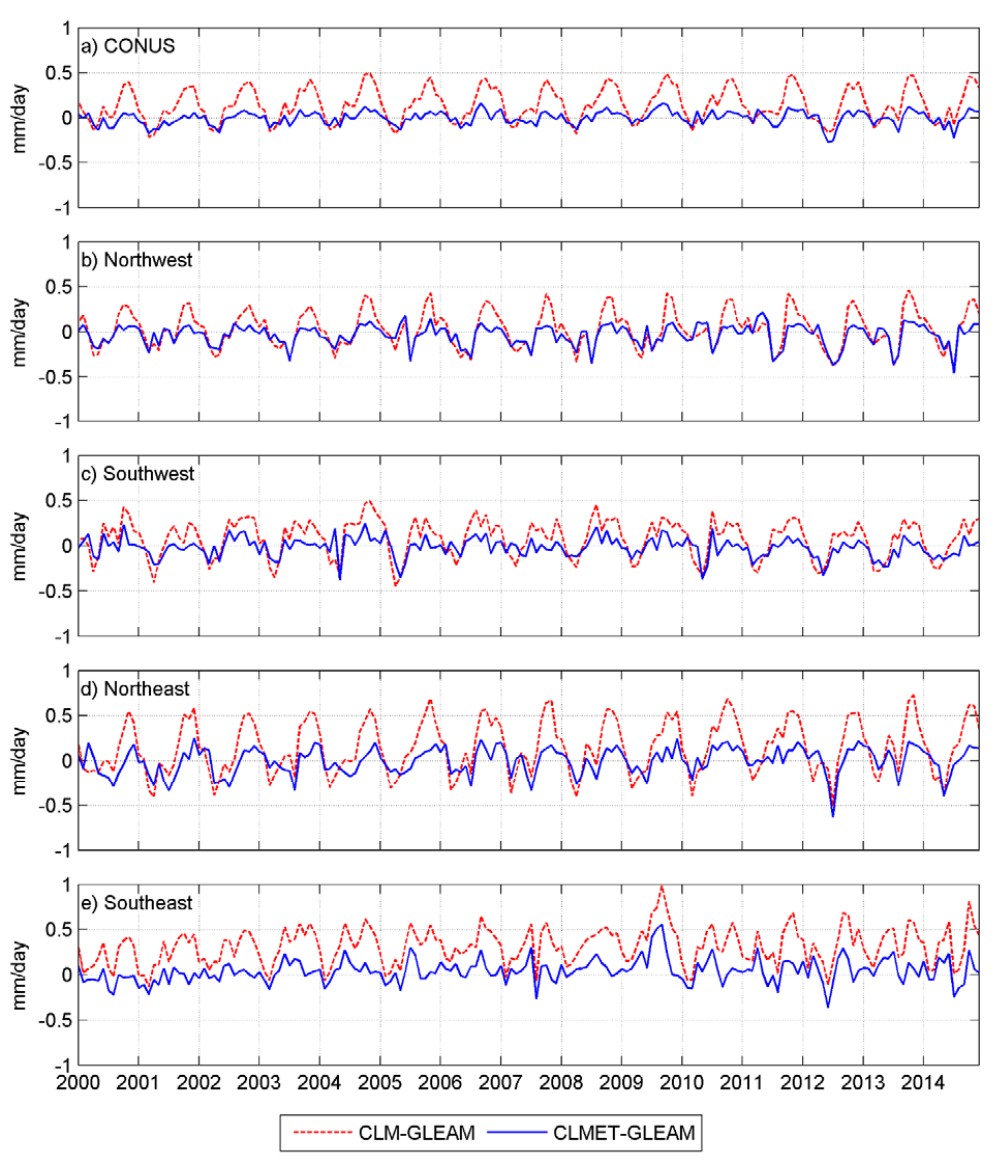


682   Figure 7 Time series of ET difference between CLM (CLMET) and GLEAM over CONUS,

683     Northwest, Southwest, Northeast, and Southeast during 2000-2014.






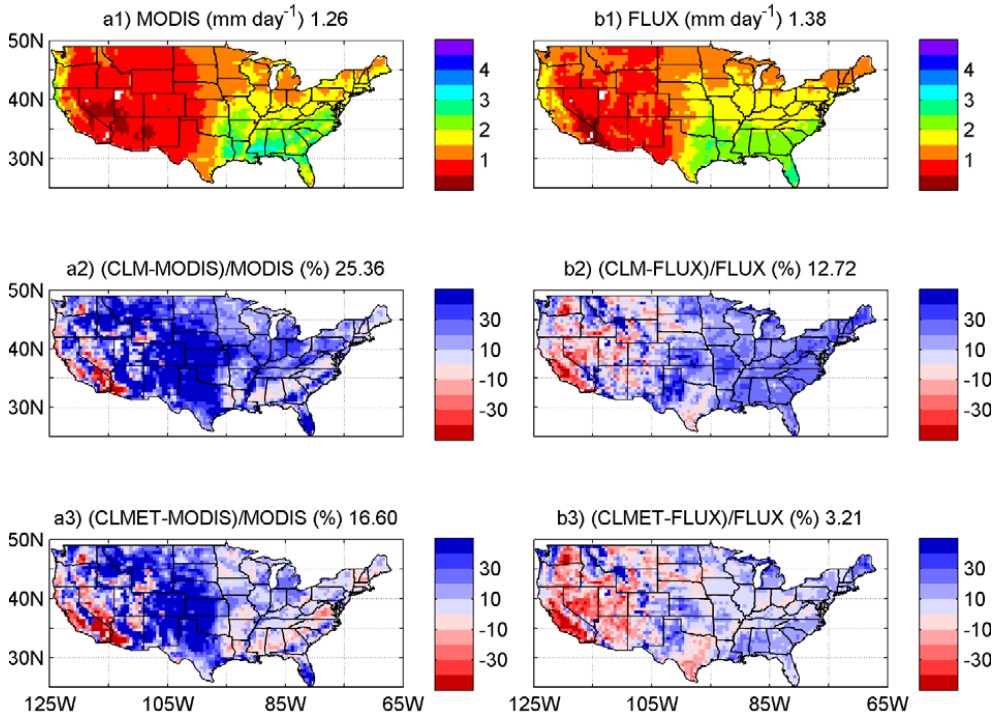


Figure 8 Mean annual ET from a) GLEAM, b) CLM, and c) CLMET, and the relative

differences between CLMET and CLM, CLM and GLEAM, and CLMET and GLEAM during

2000-2014. Numbers in titles are CONUS-averaged values.



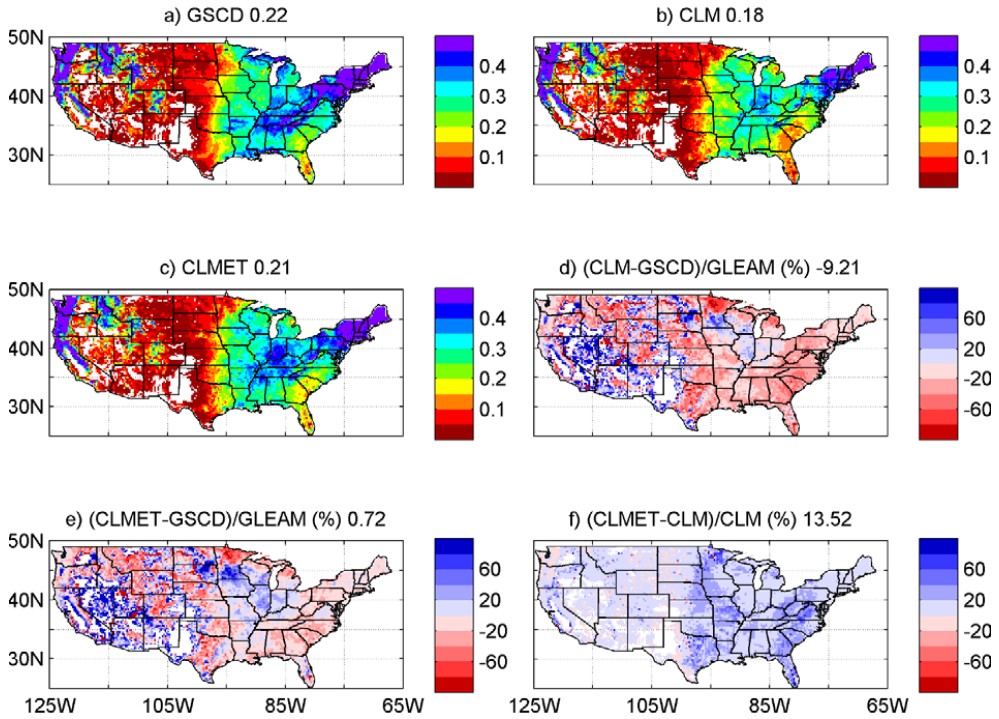


Figure 9 Mean annual runoff coefficient (the ratio runoff to total precipitation) from a) Global

Streamflow Characteristics Dataset (GSCD), b) CLM, and c) CLMET, and the relative

differences between d) CLM and GSCD, e) CLMET and GSCD, and f) CLMET and CLM

during 2000-2014. Runoff coefficient less than 0.02 is blanked out. Numbers in titles are

CONUS-averaged values.





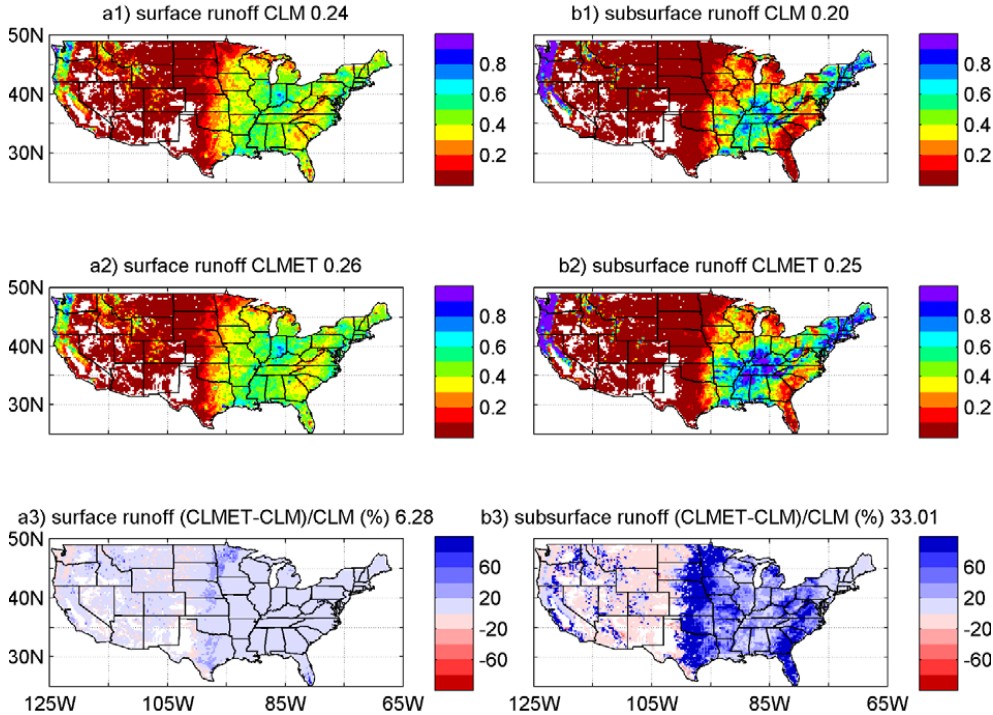


Figure 10 Surface runoff and subsurface runoff simulated in CLM and CLMET and their relative

differences during 2000-2014. Numbers in titles are the CONUS-averaged values.



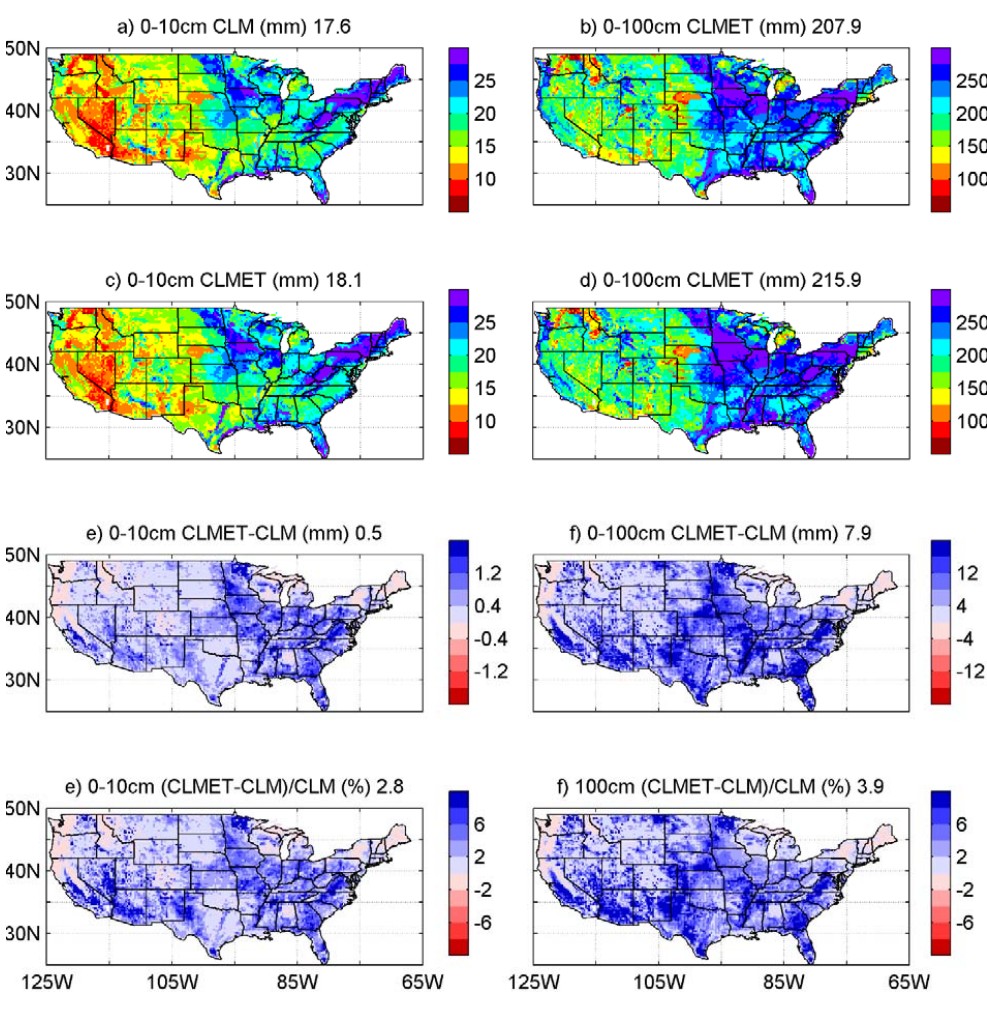

Figure 11 Simulated soil moisture (mm) in the top 0-10 cm and 0-100 layers in August from

CLM and CLMET, their differences, and their relative differences during 2000-2014.





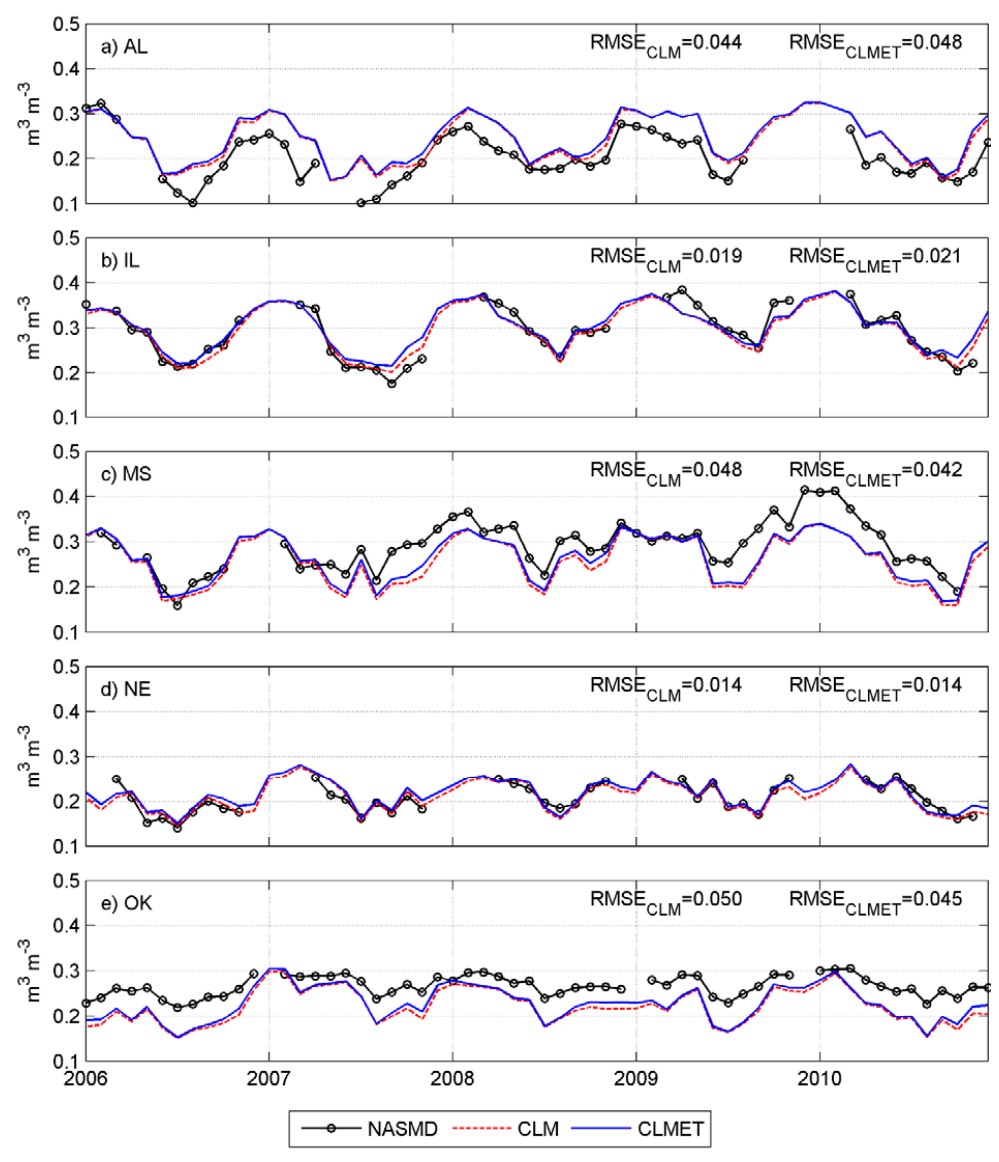


Figure 12 Monthly volumetric soil water content ($m^{-3}\,m^{-3}$) in the top 0-10cm soil layer from the

quality-controlled NASMD, CLM, and CLMET over the state of Alabama (AL), Illinois (IL),

Mississippi (MS), Nebraska (NE), and Oklahoma (OK) for the period of 2006-2010.




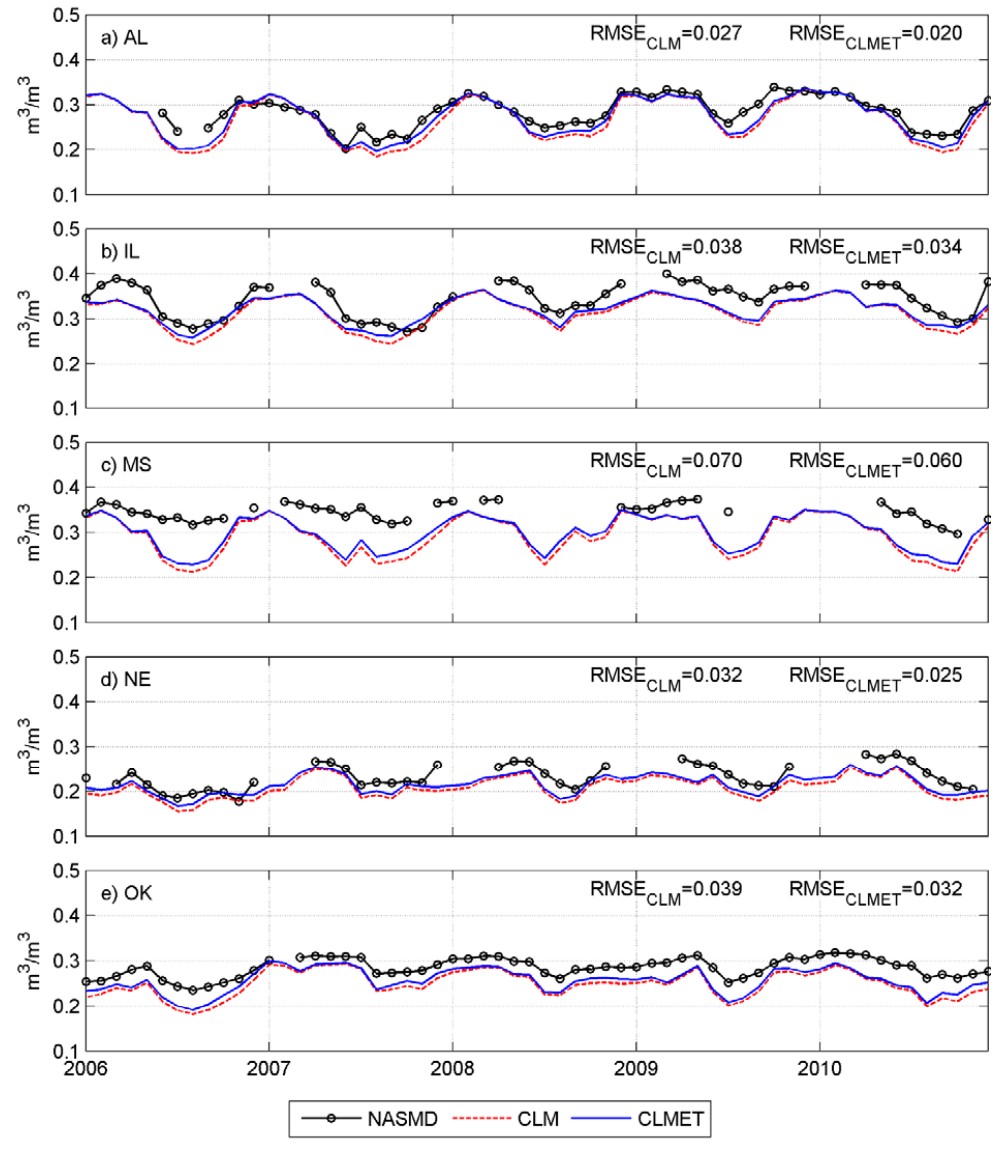


Figure 13 Same as Figure 12, but for the top 0-100cm soil layer.