# Peer review of "Incorporating remote sensing ET into Community Land Model version 4.5"

_Hydrology and Earth System Sciences, 2016_

## Referee Comment (RC2) · Anonymous Referee #2 · 2 Feb 2017

Manuscript Number: hess-2016-696 Title: Incorporating remote sensing ET into Community Land Model version 4.5 Authors: Dagang Wang, Guiling Wang, Dana T. Parr, Weilin Liao, Youlong Xia, Congsheng

Summary

This paper follows the ET bias correction scheme proposed in Parr et al. 2015 and carries out a regional scale (CONUS) study in order to evaluate the effectiveness/performance of this approach over a large domain in terms of estimating ET, runoff, and soil moisture. The main idea I see is to reduce the ET overestimation in CLM 4.5 by rescaling it down and push the reduced ET back into the model to raise the runoff and soil moisture content – this goal is obviously achieved. The data, experiments and analysis in this study are all carefully chosen and the descriptions are

very clear too. The overall quality of the research is good though most of the major conclusions are more or less well expected even without these experiments.

I think the paper can be published in HESS with minor revisions.

Major Comments

Unlike true "state" variables like moisture content or temperature, whose current value directly influences the future state of the underlying dynamic system, ET is not a state variable but a flux variable. Therefore, any effort to incorporate ET information effectively into the land surface model needs a way to propagate the change to ET flux across other parts of the dynamic system (e.g., soil moisture, canopy storage, runoff fluxes, etc.). The approach taken in this paper (following Parr et al. 2015) is to re-run the model (CLMET) and force the ET flux to be a value rescaled relative to the initial run (CLM), where the rescaling factor is pre-calibrated for every location and month. This approach is simple and effective, I think.

On the other hand, this approach is also awkward as it looks like an enhanced "post-processing" for bias correction instead of tackling the ET overestimation from its root cause, e.g., an underestimated surface resistance. The awkwardness comes in also because the "forced" ET in the CLIMET run will considerably disrupt the model physics itself, e.g. breaking the water balance and sustaining wetter soil without letting the plants transpire more. If we adjust the resistance (or some other related process like to make the water easier/faster to drain from the soil), then most of such physical inconsistency would be gone.

The authors have a major assumption that the ET biases won't change from year to year (with seasonal variability, though) so that such static errors can be corrected with static correction factors. So, the entire long ET validation section (4.2.1) is really validating the performance of the new estimation system but this stationarity assumption.

It'll be interesting if the results can be compared to a pure "post-processing" approach,

i.e., to rescale ET then rebalance the water budget between precipitation, ET, soil moisture, and runoff.

Details:

Line 65: model -> models

Line 88: intense -> intensive

Line 91: past -> historical

Line 101: Parr et al. -> Parr et al. (2015); into -> for

Line 111: spell out PFT

Line 122: "CONUS" was first mentioned in line 115

Line 155: unbalance -> imbalance

Line 322-334: where does the runoff data come from? GSCD or GRDC? What is GRDS in line 328? And Line 379?

Line 413: replace -> to replace

---

## Author Response (AR1)

We thank the two reviewers for their constructive comments. We have incorporated the review comments and revised the manuscript thoroughly. The review comments and the revision have resulted in a much more complete presentation of the work. While the changes made to the manuscript can be seen in the revised manuscript, we also present here our detailed responses to the review comments (reviewer comments in black, our response in blue).

**Responses to the comments from Reviewer #1**

GENERAL COMMENTS
The paper describes a very simple approach to attribute model biases in the simulated states and fluxes of the latest version of the Community Land Model (CLM4.5). This is an important and interesting research area, as biases in modelled soil moisture or discharge can for instance substantially affect the prediction and analysis of hydro-climatic extremes such as droughts and/or floods. The approach introduced in the paper is not really innovative as it was first published by Parr et al. in 2015; but it is tested here for a larger study area and a different land-surface model. In general, the method and the results in this paper are well-described, but–to my opinion–not really surprising and rather straightforward. Substantial parts of the results and discussions are dedicated to the differences in bias between the GLEAM-derived datasets and the CLM-runs with and without the bias correction. These results are very straightforward and predictable, as the bias-correction factors were first calibrated against GLEAM. Furthermore, most of the validations/comparisons are performed at aggregated variables (both in space and time), which might mask some of the potential issues. Summarized, I think the topic of this study is interesting, but I have the feeling that the paper (especially the results section) needs some improvements before final publication. Below I list some more specific comments.

SPECIFIC COMMENTS
1. In Section 4.2.1 it is claimed several times that the performance of CLMET is substantially better as compared to the original CLM. To my opinion, these statements need to be revised as they are not necessarily correct; especially not when the reference data is the GLEAM dataset itself. As the bias-correction factors are calculated using the GLEAM data as a reference, it makes perfect sense that applying these correction factors in the model brings the model closer to GLEAM (unless the assumption of time-invariance would not be fulfilled). Therefore, the results discussed from P11-L243 to P14-L305 (i.e. comparison of the bias-corrected CLM evaporation to the GLEAM dataset) only show the robustness of the correction factors. They do not show an improvement of CLMET in reference to CLM. To me, the evaluation of the runoff coefficients and the comparison against alternative datasets of evaporation (FLUXNETMTE, MODIS) is a step in the right direction, but only a small portion of the discussion is dedicated to these results. Therefore, I would suggest to improve the evaluation of the results to really show the impact of applying the method. I would strongly recommend to (1) validate the modelled evaporation against in situ

measurements (for instance data from single eddy-covariance towers) and, (2) extend the evaluation of the model against the alternative datasets of evaporation.

Response: We have followed the reviewer's suggestions in revising the manuscript:

1) Validate the modelled evaporation against in situ measurements:

We selected 16 eddy flux tower stations from the AmeriFlux network to validate model performance (as shown in Figure 1b of the revised manuscript). These stations were previously used to validate the NLDAS-2 surface models by Xia et al. (2015). The 16 stations are located in different sub regions of CONUS with different vegetation cover (i.e., grassland, cropland, needleleaf forest, broadleaf forest, and mixed forest). Considering both consistency in validation period and data availability, we use the year of 2005 for validation at most sites except for three sites: Sylvania Wilderness (2002), Donaldson (2004) and Walnut River (2004).

The model validations are based comparing each station with the model grid cell that encompasses the station. The station-based ET (or latent heat flux, in W/m2) are measured every 30 minutes and aggregated to daily and monthly values. Except for Port Peck and Wind River Crane stations in the northwest CONUS, for all other stations the monthly mean ET from CLMET agrees better with the observed ET than that from CLM (Figure 8 of the revised manuscript). The same statement holds for daily mean ET (Figures 9, and 10 of the revised manuscript). Generally, CLM overestimates ET as compared with station observations, and CLMET alleviates this overestimation, which is consistent with comparisons between modelled ET and satellite-based ET products.

"*In addition, the ET validation is also conducted on the site scale (Figures 8, 9, and 10).* Except for Port Peck and Wind River Crane stations in the northwest CONUS, for all other stations the monthly mean ET from CLMET agrees better with the observed ET than that from CLM *(Figure 8). The same statement holds for daily mean ET (Figures 9 and 10). Generally, CLM overestimates ET as compared with station observations, and CLMET alleviates this overestimation, which is consistent with comparisons between modelled ET and satellite-based ET products.*" (last paragraph of Section 4.2.1 in the revised manuscript)

Xia, Y., Hobbins, M. T., Mu, Q., & Ek, M. B. (2015). Evaluation of NLDAS-2 evapotranspiration against tower flux site observations. Hydrological Processes, 29(7), 1757-1771.

[Figure]

Figure 1b Locations of the 16 AmeriFlux stations with vegetation types.

[Figure]

Figure 8 Monthly mean latent heat fluxes from CLM, CLMET and observations at 16 flux tower sites. RMSECLM and RMSECLMET represent the root mean square error against observations for CLM and CLMET, respectively. Note that the CLM and CLMET simulations are driven with meteorological forcing at the grid cell level (as opposed to site-specific forcing).

[Figure]

Figure 9 Daily mean latent heat fluxes from CLM and CLMET grids and station observations at ARM SGP Burn, Audubon Grassland, Bondville, Donaldson, Flagstaff Forest, Fort Dix, Fort Peck, and Little Prospect. RMSE$_{CLM}$ and RMSE$_{CLMET}$ represent the root mean square error against observations for CLM and CLMET, respectively.

[Figure]

Figure 10 Daily mean latent heat fluxes from CLM and CLMET grids and station observations at Mead Rainfed, Metolius Pine, Missouri Ozark, Morgan Forest, Sylvania Wilderness, Tonzi Ranch, Walnut River, and Wind River Crane. RMSE$_{CLM}$ and RMSE$_{CLMET}$ represent the root mean square error against observations for CLM and CLMET, respectively.

2) extend the evaluation of the model against the alternative datasets of evaporation:

We have deleted the evaluations of ET seasonal cycle and monthly value using the GLEAM dataset, and added the evaluations using the MODIS and FLUXNET-MTE dataset. Therefore GLEAM is used for algorithm calibration while the other two ET products are used for validation. Using MODIS or FLUXNET-MTE ET as a reference,

modeled ET from CLMET is the similar to that from CLM over western CONUS, whereas CLMET substantially improves ET simulations over eastern CONUS as compared with CLM. The improvement in CLMET is more evident during September-October-November. We have added the figures (Figures 6 and 7 in the revised manuscript) and revised the relevant texts in the revised manuscript.

"*The analysis on time series of ET from MODIS, FLUXNET-MTE, and two types of simulations also demonstrates improvement from CLM to CLMET. Climatological seasonal cycles of ET over CONUS and four sub regions for 2000-2011 are shown in Figure 6. CLMET performs better than CLM over CONUS with smaller RMSE (0.31 versus 0.40 against MODIS, 0.19 versus 0.25 against FLUXNET-MTE). The improvement mainly results from reduction of overestimation existing in CLM for SON and DJF. However, the model performance greatly varies with region. As indicated by the ET RMSE values, CLMET and CLM perform similarly over western CONUS, whereas CLMET improves the ET simulation over eastern CONUS no matter which reference data is used.    Figure 7 compares the temporal evolution of the simulated ET in CLM and CLMET against MODIS and FLUXNET-MTE ET over CONUS and four sub-regions. It is evident that the bias correction method in CLMET is very effective in reducing overestimation (positive bias), but does not work as well in correcting the underestimation (negative bias). The difference has to do with the specific ET regime, i.e. whether ET is limited by water or energy.    When an overestimated ET is overwritten with a lower value, the water on land is sufficient to support the reduced ET; in contrast, when an underestimate ET is overwritten with a higher value, the land surface model checks whether water storage in soil layer and vegetation canopy can sustain the elevated ET and further adjust if necessary to keep with the mass conservation equation. The extent to which ET increases is limited by the availability of water stored in soil layer and vegetation canopy. Therefore, in case of water-limited ET, the actual ET after the water availability check in CLMET can be substantially lower than the corrected ET fed into model.*" (the second paragraph from bottom of Section 4.2.1 in the revised manuscript)

[Figure]

Figure 6 Seasonal cycles of ET from MODIS, FLUXNET-MTE, CLM, and CLMET over CONUS, Northwest, Southwest, Northeast, and Southeast during the period 2000-2011.

[Figure]

Figure 7 Time series of ET difference between model (CLM or CLMET) and reference data (MODIS or FLUXNET-MTE) over CONUS, Northwest, Southwest, Northeast, and Southeast during the period 2000-2011.

2. It is not clear to me how the statistics in Tables 1 to 4 are exactly calculated. This should be better documented in the manuscript. For instance, the temporal statistics in Table 2: are these calculated per pixel and subsequently averaged over the different study areas (CONUS, NW ...)? Or is the modelled evaporation first aggregated for the study area, and the statistics calculated on the aggregated values? In addition, next to the comparison against the FLUXNET-MTE product, I would also suggest to at least include a validation of the products against actual FLUXNET measurements. Although there are different issues with eddy-covariance measurements as well, a lot of data is

freely available, and these measurements are probably closer to the truth than any of the datasets currently used in the study.

Response:

1) On the calculation of the statistics in Tables 1 to 4:

We have added the following equations to the revised manuscript to show how the statistics is calculated.

$$Bias = \frac{1}{N}\sum_{i=1}^{i=N}\left(\overline{S_i} - \overline{R_i}\right)$$

$$Relative\ bias = \frac{1}{N}\sum_{i=1}^{i=N}\frac{\left(\overline{S_i} - \overline{R_i}\right)}{R_i}$$

$$RMSE = \sqrt{\frac{\sum_{i=1}^{i=N}\left(\overline{S_i} - \overline{R_i}\right)^2}{N}}$$

Where N is the total number of grid cells, and $\overline{S_i}$ ($\overline{R_i}$) are the temporal average of model simulated (reference) value for grid cell i, which is calculated as:

$$\overline{S_i} = \frac{1}{M}\sum_{j=1}^{j=M} S_{i,j}$$

$$\overline{R_i} = \frac{1}{M}\sum_{j=1}^{j=M} R_{i,j}$$

Where Si,j (Ri,j) is model simulated (reference) value on time j and at grid cell i, M is the total number of time series. The statistic RMSE is also used to validate models in reproducing temporal series where M becomes the total number of grid cells, and N becomes the total number of time series.

*"In this study, the statistics Bias, Relative bias, and root mean square error (RMSE) are used to validate models in reproducing the spatial pattern against the reference dataset. They are defined as:*

$$Bias = \frac{1}{N}\sum_{i=1}^{i=N}\left(\overline{S_i} - \overline{R_i}\right) \tag{1}$$

$$Relative\ bias = \frac{1}{N}\sum_{i=1}^{i=N}\frac{\left(\overline{S_i} - \overline{R_i}\right)}{R_i} \tag{2}$$

$$RMSE = \sqrt{\frac{\sum_{i=1}^{i=N}\left(\overline{S_i} - \overline{R_i}\right)^2}{N}} \tag{3}$$

*Where N is the total number of grid cells, and $\overline{S_i}$ ($\overline{R_i}$) are the temporal average of model simulated (reference) value for grid cell i, which is calculated as:*

$$\overline{S_i} = \frac{1}{M} \sum_{j=1}^{j=M} S_{i,j} \qquad\qquad (4)$$

$$\overline{R_i} = \frac{1}{M} \sum_{j=1}^{j=M} R_{i,j} \qquad\qquad (5)$$

*Where Si,j (Ri,j) is model simulated (reference) value on time j and at grid cell i, M is the total number of time series. The statistic RMSE is also used to validate models in reproducing temporal series where M becomes the total number of grid cells, and N becomes the total number of time series." (the last two paragraphs of Section 2.2)*

2) validation of the products against actual FLUXNET measurements
    We have added validations against actual FLUXNET measurements at 16 stations. Please see the response to comment 1 for details.

3. I have the feeling that some issues of the method (e.g. the assumption of time invariant scaling factors or the use of monthly scaling factors) might be masked by the spatiotemporal scales at which the results are analyzed. For instance, why are only time series of the climatological cycle for the entire study area shown in Figure 6? It could be interesting to show some time series from individual pixels as well. Also, an analysis at shorter time scales might show some interesting results. E.g. why do the authors not show a time series of daily evaporation? The same holds for Figure 12: why are these time series not shown at daily time steps and on a pixel basis?
Response:
1) daily series of ET for individual pixels
    We have included ET evaluation on daily and monthly scales at 16 pixels, and added figures to compare model simulations with in-situ observations. Please see the response to comment 1 for details.
2) daily series of soil moisture for individual pixels
    It is difficult to determine which sites are suitable for validation from total 232 soil moisture observation sites. And the comparison between model simulations and site observations on the daily scale is consistent with the comparison on the monthly scale, as indicated by the comparison for ET. Therefore, we decide to still keep figures on the comparison at the state level (Figures 14 and 15).

4. P6-L116-117: Could the authors be more specific here about what is meant by spatial correlation? Observations from FLUXNET are essentially point measurements. How are spatial correlations defined here?
Response: Parr et al. (2016) used FLUXNET-MTE (model tree ensemble) ET, which is a gridded ET product, to evaluate CLM4.5. We changed the description as follows:

*The spatial correlation coefficients between the simulated annual ET and the FLUXNET-MTE (model tree ensemble) ET are as high as 0.93.*

5. P5-L107: I think it should be mentioned here at what temporal resolution the model is applied. From the results in Table 2, I can guess the model is run at a daily resolution. If the latter is the case, I think it should also be justified why the scaling factors are calculated at the monthly time scale. Given that both the simulations and the GLEAM datasets are available at a daily resolution, the scaling factors could as well be calculated at the daily scale. Would this also work? Did the authors test the effect of applying daily scaling factors in the algorithm?

Response:

1) temporal resolution of model: the temporal resolution of model is one hour, which is typical for land surface models. We have added this information into Section 2.3 of the revised manuscript.

2) temporal resolution of scaling factor: This scaling factor characterize the relationship between model biases and ET climatology, and the fundamental assumption is that the nature of the model biases is time-invariant at the inter-annual and longer time scales. The monthly time scale is used here to account for its seasonality. To say that the nature of the model biases varies on a day-to-day time scale does not make physical sense, although technically it can be done. In fact we tested the performance of CLMET based on daily scaling factors. CLMET performance is not improved using daily scaling factors as compared with CLMET using monthly scaling factors.

6. P11-L244-245: Please revise this sentence: GLEAM data is not missing in this period, but is probably masked out in this study as the Northern regions of CONUS are typically covered with snow during these times of the year. GLEAM estimates of sublimation are available for these regions, but I guess they are not considered here (at P7-L140-141, it reads that only interception loss, transpiration and bare-soil evaporation are considered).

Response: we deleted the data records in some parts of west CONUS during the cold seasons by mistake, when GLEAM-derived ET is negative. This led to many missing values in annual ET map in Figure 4 of the original manuscript. We have corrected this mistake and updated Figure 4 (Figure 3 in the revised manuscript). The CONUS-averaged value from CLMET in the update version of annual ET is slightly better than the value in the previous version. We have also updated the table 1 to reflect this change.

[Figure]

Figure 3 Mean annual ET from a) GLEAM, b) CLM, and c) CLMET, and the relative differences between d) CLM and GLEAM, e) CLMET and GLEAM, and f) CLMET and CLM during 2000-2014. Numbers in titles are CONUS-averaged values.

7. P12-L261-262: If the term "significant" is used, it implies that a statistical test was applied to check this hypothesis. If this is the case, the test should be mentioned here. Response: we changed to "substantially".

8. Please note that the GLEAM datasets are no "observations" of evaporation. They are estimates of terrestrial evaporation, resulting from applying a simple conceptual model to observation-based datasets of different meteorological variables. GLEAM is kept as simple as possible to minimize the impact of the algorithms and maximize the impact of the meteorological observations on the estimates of evaporation. I would suggest to revise this throughout the manuscript.
Response: we have changed from "observations" to "estimations".

TECHNICAL CORRECTIONS
1. Please use hyphens in "compound adjectives" such as "land-surface models" or "widely-used tools".
Response: the expression of "land surface models" and "widely used tools" are widely used in literature.

2. I would suggest explaining all abbreviations upon their first use. E.g. P3-L68-69: SAC and VIC.

Response: Following reviewer's suggestion, we have spelled out SAC-SMA (Sacramento Soil Moisture Accounting) and VIC (Variable Infiltration Capacity) when they appeared for the first time in the revised manuscript.

"*The Mosaic and Sacramento Soil Moisture Accounting (SAC-SMA) models tend to overestimate ET, whereas the Noah and Variable Infiltration Capacity (VIC) models are likely to underestimate ET.*" (the last sentence of first paragraph, Section 1)

3. P5-L108: Given that no further details are provided in the paper regarding the land surface model used, I would suggest adding a reference here for the CLM model.
Response: Following reviewer's suggestion, we have added a reference about Community Land Model version 4.5 when the model is introduced in the revised manuscript.

Oleson, K. W. et al.: Technical Description of version 4.5 of the Community Land Model (CLM), NCAR Tech. Note, NCAR/TN-503+STR, doi:10.5065/D6RR1W7M, 2013.

4. P5-L111: Please define "PFT".
Response:   Defined

5. P6-L124: I guess this should be section 2.2 instead of 2.3.
Response: Yes, it is 2.2. We have corrected it.

6. P7-L161: The fact that the GLEAM database has three subsets is not relevant here if you only use one.
Response: Following reviewer's suggestion, we have deleted the description of three subsets of GLEAM.

7. P28-Table1: Please correct "COUNS" in the caption. Please also check this at other places in the manuscript: e.g. P14-L315
Response: We have corrected them to "CONUS".

8. P34-Figure3: Please explain in the caption which areas are masked. I guess these are regions covered with snow?
Response: The GLEAM-derived dew and the CLM simulated dew is not consistent in some areas of northwest CONUS. If that happens, the scaling factors became negative, because ET is negative for one and positive for the other. We did not scale ET when the scaling factor is negative, and those areas are masked out in Figure 2. We have added an explanation about it.

**Responses to the comments from Reviewer #2**

Manuscript Number: hess-2016-696 Title: Incorporating remote sensing ET into Community

Land Model version 4.5 Authors: Dagang Wang, Guiling Wang, Dana T. Parr,

Weilin Liao, Youlong Xia, Congsheng

Summary

This paper follows the ET bias correction scheme proposed in Parr et al. 2015 and carries out a regional scale (CONUS) study in order to evaluate the effectiveness/performance of this approach over a large domain in terms of estimating ET, runoff, and soil moisture. The main idea I see is to reduce the ET overestimation in CLM 4.5 by rescaling it down and push the reduced ET back into the model to raise the runoff and soil moisture content – this goal is obviously achieved. The data, experiments and analysis in this study are all carefully chosen and the descriptions are very clear too. The overall quality of the research is good though most of the major conclusions are more or less well expected even without these experiments.

I think the paper can be published in HESS with minor revisions.

Major Comments

Unlike true "state" variables like moisture content or temperature, whose current value directly influences the future state of the underlying dynamic system, ET is not a state variable but a flux variable. Therefore, any effort to incorporate ET information effectively into the land surface model needs a way to propagate the change to ET flux across other parts of the dynamic system (e.g., soil moisture, canopy storage, runoff fluxes, etc.). The approach taken in this paper (following Parr et al. 2015) is to re-run the model (CLMET) and force the ET flux to be a value rescaled relative to the initial run (CLM), where the rescaling factor is pre-calibrated for every location and month. This approach is simple and effective, I think. On the other hand, this approach is also awkward as it looks like an enhanced post-processing" for bias correction instead of tackling the ET overestimation from its root cause, e.g., an underestimated surface resistance. The awkwardness comes in also because the "forced" ET in the CLIMET run will considerably disrupt the model physics itself, e.g. breaking the water balance and sustaining wetter soil without letting the plants transpire more. If we adjust the resistance (or some other related process like to make the water easier/faster to drain from the soil), then most of such physical inconsistency would be gone.

Response: the model bias in ET simulations results from inaccurate information of meteorological conditions (Mueller and Seneviratne, 2014), surface-type data (Hwang and Choi, 2013), model parameters (Ma et al. 2015), and soil water (Decker 2015). Adjusting surface resistance is essentially one of many methods of model parameters calibration, which can reduces model bias as well. However, only making parameter adjustment may results in nonphysical parameter subsets when other inaccurate information is the main cause of the model bias for some regions/seasons (Ray et al. 2015). In this study, we take a different approach to correct simulated ET as a whole instead of adjusting each separate factors, which provide a simple and efficient way to

improve model performance in hydrological estimation without improving the model physics itself. We have added a short discussion in the Section 5.

*"Model parameter calibration (e.g., tuning surface resistance) is another way to reduce model bias (Ren et al. 2016). However, the parameter space may contain nonphysical parameter subsets (Ray et al. 2015), which is especially an issue when model parameter tuning is used to offset unrelated model deficits. The method used in this study attempts to avoid such issues through improving the model performance without dealing with calibration of model physical parameters."* (the last pargragrah of Section 5 in the revised manuscript)

Mueller, B., and S. I. Seneviratne (2014), Systematic land climate and evapotranspiration biases in CMIP5 simulations, Geophysical Research Letter, 41, 128–134, doi:10.1002/2013GL058055.

Hwang, K., and Choi, M. (2013). Seasonal trends of satellite-based evapotranspiration algorithms over a complex ecosystem in East Asia. Remote Sensing of Environment, 244-263.

Ma, N., Y. Zhang, C.-Y. Xu, and J. Szilagyi (2015), Modeling actual evapotranspiration with routine meteorological variables in the data-scarce region of the Tibetan Plateau: Comparisons and implications, Journal Geophysical Research: Biogeosciences, 120, doi:10.1002/2015JG003006.

Decker, M. (2015). Development and evaluation of a new soil moisture and runoff parameterization for the CABLE LSM including subgrid-scale processes. Journal of Advances in Modeling Earth Systems, 7(4), 1788-1809.

Ray, J., Z. Hou, M. Huang, K. Sargsyan, and L. Swiler (2015), Bayesian calibration of the Community Land Model using surrogates, SIAM/ASA Journal on Uncertainty Quantification, 199–233, doi:10.1137/140957998.

The authors have a major assumption that the ET biases won't change from year to year (with seasonal variability, though) so that such static errors can be corrected with static correction factors. So, the entire long ET validation section (4.2.1) is really validating the performance of the new estimation system but this stationarity assumption. It'll be interesting if the results can be compared to a pure "post-processing" approach, i.e., to rescale ET then rebalance the water budget between precipitation, ET, soil moisture, and runoff.

Response: It is hard to rebalance water and energy budgets though post processing without model runs after ET is rescaled. The rescaled ET influences simulations of many components of land surface processes, such as infiltration, soil water/energy transport, which cause changes in land surface states. The land surface states at the current time step is the bases of flux variable simulations for the next time step. All

these processes and connections between adjacent time steps cannot be tackled in the post processing. To obtain the consistency between different components of land surface processes and connect land surface states between adjacent time steps, we really need to re-run CLM and let model resolve all these issues. That is the reason why Parr et al. (2015) proposed the method and we applied this method in CLM on the regional scale.

Details:
Line 65: model -> models
Response: we have changed to "models".

Line 88: intense -> intensive
Response: we have changed to "intensive".

Line 91: past -> historical
Response: we have changed to "historical".

Line 101: Parr et al. -> Parr et al. (2015); into -> for
Response: we have changed to "Parr et al. (2015)" and "for".

Line 111: spell out PFT
Response: we have spelled out PFT (plant functional type).

Line 122: "CONUS" was first mentioned in line 115
Response: we have define "CONUS" (Conterminous United States) in line 115.

Line 155: unbalance -> imbalance
Response: we have changed to "imbalance".

Line 322-334: where does the runoff data come from? GSCD or GRDC? What is GRDS in line 328? And Line 379?
Response: all these should be GSCD (Global Streamflow Characteristics Dataset). We have corrected them.

Line 413: replace -> to replace
Response: we have changed to "to replace".

**Incorporating remote sensing ET into Community Land Model version 4.5**

Dagang Wang[1, 2, 3, 4]*, Guiling Wang[4]*, Dana T. Parr[4], Weilin Liao[1, 2], Youlong Xia[5], Congsheng Fu[4]

[1] School of Geography and Planning, Sun Yat-sen University, Guangzhou, China

[2] Guangdong Key Laboratory for Urbanization and Geo-simulation, Sun Yat-sen University, Guangzhou, China

[3] Key Laboratory of Water Cycle and Water Security in Southern China of Guangdong High Education Institute, Sun Yat-sen University, Guangzhou, P.R. China

[4] Department of Civil and Environmental Engineering, University of Connecticut, Storrs, USA

[5] National Centers for Environmental Prediction/Environmental Modeling Center, and I. M. System Group at NCEP/EMC, College Park, Maryland, USA

Submitted to special issue of Hydrology and Earth System Sciences: Observations and modeling of land surface water and energy exchanges across scales, in Honor of Eric F. Wood

Revised April, 2017

[revised manuscript text omitted]

In this study, the statistics Bias, Relative bias, and root mean square error (RMSE) are used to validate models in reproducing the spatial pattern against the reference dataset. They are defined as:

$$Bias = \frac{1}{N}\sum_{i=1}^{i=N}\left(\overline{S_i} - \overline{R_i}\right) \tag{1}$$

$$Relative\ bias = \frac{1}{N}\sum_{i=1}^{i=N}\frac{\left(\overline{S_i} - \overline{R_i}\right)}{R_i} \tag{2}$$

$$RMSE = \sqrt{\frac{\sum_{i=1}^{i=N}\left(\overline{S_i} - \overline{R_i}\right)^2}{N}} \tag{3}$$

Where N is the total number of grid cells, and $\overline{S_i}$ ( $\overline{R_i}$ ) are the temporal average of model simulated (reference) value for grid cell i, which is calculated as:

$$\overline{S_i} = \frac{1}{M}\sum_{j=1}^{j=M} S_{i,j} \tag{4}$$

$$\overline{R_i} = \frac{1}{M}\sum_{j=1}^{j=M} R_{i,j} \tag{5}$$

Where $S_{i,j}$ ($R_{i,j}$) is model simulated (reference) value on time j and at grid cell i, M is the total number of time series. The statistic RMSE is also used to validate models in reproducing temporal series where M becomes the total number of grid cells, and N becomes the total number of time series.

**3 Data**

3.1 ET

3.1.1 GLEAM ET

[revised manuscript text omitted]

other three seasons except for MAM (Tables 2 and 3). Bias, relative bias and RMSE in CLMET is greater than CLM for the whole CNOUS, Northwest, Southwest, and Northeast in MAM. Among all other three seasons, SON is the reason when model performance is improved most from CLM to CLMET. The performance in CLMET against MODIS or FLUXNET-MTE is similar to the model performance against GLEAM for annual mean, JJA, SON, and DJF but with smaller magnitudes. CLMET deteriorates the ET simulation for MAM by intensifying overestimation already occurring in CLM, which is different from the validation against the GLEAM-based ET.

The analysis on time series of ET from MODIS, FLUXNET-MTE, and two types of simulations also demonstrates improvement from CLM to CLMET. Climatological seasonal cycles of ET over CONUS and four sub regions for 2000-2011 are shown in Figure 6. CLMET performs better than CLM over CONUS with smaller RMSE (0.31 versus 0.40 against MODIS, 0.19 versus 0.25 against FLUXNET-MTE). The improvement mainly results from reduction of overestimation existing in CLM for SON and DJF. However, the model performance greatly varies with region. As indicated by the ET RMSE values, CLMET and CLM perform similarly over western CONUS, whereas CLMET improves the ET simulation over eastern CONUS no matter which reference data is used. Figure 7 compares the temporal evolution of the simulated ET in CLM and CLMET against MODIS and FLUXNET-MTE ET over CONUS and four sub-regions. It is evident that the bias correction method in CLMET is very effective in reducing overestimation (positive bias), but does not work as well in correcting the underestimation (negative bias). The difference has to do with the specific ET regime, i.e. whether ET is limited by water or energy. When an overestimated ET is overwritten with a lower value, the water on land is sufficient to support the reduced ET; in contrast, when an underestimate ET is overwritten with a higher value, the land surface model checks whether water storage in soil layer and vegetation canopy can

sustain the elevated ET and further adjust if necessary to keep with the mass conservation equation. The extent to which ET increases is limited by the availability of water stored in soil layer and vegetation canopy. Therefore, in case of water-limited ET, the actual ET after the water availability check in CLMET can be substantially lower than the corrected ET fed into model.

In addition, the ET validation is also conducted on the site scale (Figures 8, 9, and 10). Except for Port Peck and Wind River Crane stations in the northwest CONUS, for all other stations the monthly mean ET from CLMET agrees better with the observed ET than that from CLM (Figure 8). The same statement holds for daily mean ET (Figures 9 and 10). Generally, CLM overestimates ET as compared with station observations, and CLMET alleviates this overestimation, which is consistent with comparisons between modelled ET and satellite-based ET products.

4.2.2 Runoff

[revised manuscript text omitted]

Oleson, K. W., Lawrence, D. M., Bonan, G. B., Drewniak, B., Huang, M., Koven, C. D., Levis, S., Li, F., Riley, W. J., Subin, Z. M., Swenson, S. C., Thornton, P. E., Bozbiyik, A., Fisher,

R. A., Kluzek, E., Lamarque, J.-F., Lawrence, P. J., Leung, L. R., Lipscomb, W., Muszala, S., Ricciuto, D. M., Sacks, W. J., Sun, Y., Tang, J. Y., and Yang, Z.-L.: Technical Description of version 4.5 of the Community Land Model (CLM), NCAR Tech. Note, NCAR/TN-503+STR, doi:10.5065/D6RR1W7M, 2013.

Parr, D., Wang, G., and Bjerklie, D.: Integrating Remote Sensing Data on Evapotranspiration and Leaf Area Index with Hydrological Modeling: Impacts on Model Performance and Future Predictions, J. Hydrometeorol., 16, 2086-2100, 2015.

Parr, D., Wang, G., and Fu, C.: Understanding Evapotranspiration Trends and their Driving Mechanisms over the NLDAS Domain Based on Numerical Experiments Using CLM4.5, Journal of Geophysical Research Atmospheres, 121, 7729-7745, 2016.

Ray, J., Z. Hou, M. Huang, K. Sargsyan, and L. Swiler: Bayesian calibration of the Community Land Model using surrogates, SIAM/ASA Journal on Uncertainty Quantification, 199–233, 2015.

Reichle, R. H. and Koster, R. D.: Global assimilation of satellite surface soil moisture retrievals into the NASA Catchment land surface model, Geophys. Res. Lett., 32, 177-202, 2005.

Ren, H., Z. Hou, M. Huang, J. Bao, Y. Sun, T. Tesfa, and R. Leung: Classification of hydrological parameter sensitivity and evaluation of parameter transferability across 431 US MOPEX basins, J. Hydrol., 536, 92–108, 2016.

Rodell, M., Houser, P. R., Jambor, U., Gottschalck, J. C., Mitchell, K., Meng, C. J., Arsenault, K. R., Cosgrove, B. A., Radakovich, J., Bosilovich, M. G., Entin, J. K., Walker, J. P., Lohmann, D., and Toll, D. L.: The Global Land Data Assimilation System, B. Am. Meteorol. Soc., 85, 381-394, 2004.

Sheffield, J. and Wood, E. F.: Characteristics of global and regional drought, 1950-2000: Analysis of soil moisture data from off-line simulation of the terrestrial hydrologic cycle, Journal of Geophysical Research Atmospheres, 112, D17115, 2007.

Spennemann, P. C. and Saulo, A. C.: An estimation of the land-atmosphere coupling strength in South America using the Global Land Data Assimilation System, Int. J. Climatol., 35, 4151-4166, 2015.

Swenson, S. C. and Lawrence, D. M.: A GRACE‐based assessment of interannual groundwater dynamics in the Community Land Model, Water Resour. Res., 51, 8817-8833, 2015.

Syed, T. H., Famiglietti, J. S., Rodell, M., Chen, J., and Wilson, C. R.: Analysis of terrestrial water storage changes from GRACE and GLDAS, Water Resour. Res., 44, 339-356, 2008.

Ukkola, A. M., Kauwe, M. G. D., Pitman, A. J., Best, M. J., Abramowitz, G., Haverd, V., Decker, M., and Haughton, N.: Land surface models systematically overestimate the intensity, duration and magnitude of seasonal-scale evaporative droughts, Environ. Res. Lett., 11, 2016.

Wang, A., Zeng, X., and Guo, D.: Estimates of global surface hydrology and heat fluxes from the Community Land Model (CLM4.5) with four atmospheric forcing datasets, J. Hydrometeorol., 17, 2493-2510, 2016.

Xia, Y., Cosgrove, B. A., Mitchell, K. E., Peters Lidard, C. D., Ek, M. B., Kumar, S., Mocko, D., and Wei, H.: Basin‐scale assessment of the land surface energy budget in the National Centers for Environmental Prediction operational and research NLDAS-2 systems, J. Geophys. Res., 121, 196-220, 2016a.

Xia, Y., Ford, T. W., Wu, Y., Quiring, S. M., and Ek, M. B.: Automated Quality Control of In Situ Soil Moisture from the North American Soil Moisture Database Using NLDAS-2 Products, Journal of Applied Meteorology & Climatology, 54, 2015a.

Xia, Y., Hobbins, M. T., Mu, Q., & Ek, M. B.  Evaluation of NLDAS-2 evapotranspiration against tower flux site observations. Hydrological Processes, 29(7), 1757-1771, 2015b.

Xia, Y., Mitchell, K. E., Ek, M. B., Cosgrove, B., Sheffield, J., Luo, L., Alonge, C., Wei, H., Meng, J., Livneh, B., Duan, Q., and Lohmann, D.: Continental-scale water and energy flux analysis and validation for North American Land Data Assimilation System project phase 2 (NLDAS‐2): 2. Validation of model‐simulated streamflow, J. Geophys. Res., 117, D3110, 2012a.

Xia, Y., Mitchell, K., Ek, M., Sheffield, J., Cosgrove, B., Wood, E., Luo, L., Alonge, C., Wei, H., Meng, J., Livneh, B., Lettenmaier, D., Koren, V., Duan, Q., Mo, K., Fan, Y., and Mocko, D.: Continental-scale water and energy flux analysis and validation for the North American Land Data Assimilation System project phase 2 (NLDAS-2): 1. Intercomparison and application of model products, Journal of Geophysical Research Atmospheres, 117, D3109, 2012b.

Xia, Y., Peters-Lidard, C. D., and Luo, L.: Basin-scale assessment of the land surface water budget in the National Centers for Environmental Prediction operational and research NLDAS-2 systems, J. Geophys. Res., 121, 196-220, 2016b.

Yin, J., Zhan, X., Zheng, Y., Liu, J., Fang, L., and Hain, C. R.: Enhancing Model Skill by Assimilating SMOPS Blended Soil Moisture Product into Noah Land Surface Model, J. Hydrometeorol., 16, 917-931, 2015c.

Table 1 Spatial evaluations of simulated ET from two different types of runs (CLM and CLMET) against GLEAM-derived ET over CONUS, Northwest (NW), Southwest (SW), Northeast (NW), and Southeast (SW) annually and seasonally during the period 2000-2014. March-April-May: MAM, June-July-August: JJA, September-October-November: SON, December-January-February: DJF

| Season | Region | Bias (mm day$^{-1}$) | | Relative bias (%) | | RMSE (mm day$^{-1}$) | |
|--------|--------|------|-------|------|-------|------|-------|
| | | CLM | CLMET | CLM | CLMET | CLM | CLMET |
| Annual | CONUS | 0.137 | -0.006 | 10.8 | -0.1 | 0.266 | 0.144 |
| | NW | 0.029 | -0.03 | 7.9 | 0.3 | 0.25 | 0.199 |
| | SW | 0.074 | -0.025 | 10.2 | -3.1 | 0.181 | 0.118 |
| | NE | 0.138 | -0.012 | 9.6 | -0.1 | 0.243 | 0.132 |
| | SE | 0.315 | 0.041 | 15.6 | 2.1 | 0.355 | 0.099 |
| MAM | CONUS | -0.081 | -0.062 | -5.8 | -3.3 | 0.351 | 0.227 |
| | NW | -0.138 | -0.074 | -6.7 | -2.7 | 0.326 | 0.244 |
| | SW | -0.211 | -0.122 | -17.9 | -9.3 | 0.318 | 0.206 |
| | NE | -0.191 | -0.078 | -8.3 | -2.8 | 0.429 | 0.293 |
| | SE | 0.19 | 0.023 | 8.9 | 1.5 | 0.346 | 0.165 |
| JJA | CONUS | 0.094 | -0.041 | 6.4 | -1.3 | 0.451 | 0.331 |
| | NW | -0.137 | -0.121 | -3.9 | -4.0 | 0.487 | 0.408 |
| | SW | 0.147 | -0.006 | 18.3 | -0.9 | 0.352 | 0.232 |
| | NE | 0.045 | -0.124 | 2.5 | -2.7 | 0.55 | 0.452 |
| | SE | 0.332 | 0.075 | 9.1 | 2.1 | 0.414 | 0.181 |
| SON | CONUS | 0.360 | 0.055 | 51 | 7.8 | 0.428 | 0.155 |
| | NW | 0.271 | 0.044 | 76.4 | 14.0 | 0.346 | 0.147 |
| | SW | 0.228 | 0.044 | 39.5 | 5.0 | 0.282 | 0.117 |
| | NE | 0.481 | 0.077 | 50.4 | 7.3 | 0.527 | 0.242 |
| | SE | 0.499 | 0.061 | 34.5 | 4.1 | 0.531 | 0.11 |
| DJF | CONUS | 0.182 | 0.009 | 77.7 | 18.9 | 0.265 | 0.115 |
| | NW | 0.114 | -0.013 | 104.2 | 28.8 | 0.252 | 0.122 |
| | SW | 0.132 | -0.014 | 42.3 | -1.9 | 0.182 | 0.056 |
| | NE | 0.239 | 0.077 | 146.4 | 65.3 | 0.334 | 0.199 |
| | SE | 0.24 | 0.004 | 49.5 | 2.7 | 0.292 | 0.072 |

Table 2. Similar to Table 1, but based on comparison with MODIS-derived ET during the period 2000-2011.

| Season | Region | Bias (mm day$^{-1}$) | | Relative bias (%) | | RMSE (mm day$^{-1}$) | |
|---|---|---|---|---|---|---|---|
| | | CLM | CLMET | CLM | CLMET | CLM | CLMET |
| Annual | CONUS | 0.321 | 0.177 | 30.8 | 19.1 | 0.427 | 0.321 |
| | NW | 0.28 | 0.232 | 35.8 | 27.9 | 0.367 | 0.326 |
| | SW | 0.282 | 0.183 | 39.7 | 25.6 | 0.428 | 0.36 |
| | NE | 0.278 | 0.125 | 19.6 | 9.1 | 0.316 | 0.193 |
| | SE | 0.431 | 0.159 | 24.9 | 10.6 | 0.538 | 0.348 |
| MAM | CONUS | 0.514 | 0.533 | 50.1 | 55.8 | 0.631 | 0.635 |
| | NW | 0.564 | 0.628 | 67.2 | 74.5 | 0.636 | 0.687 |
| | SW | 0.345 | 0.438 | 45.9 | 61.8 | 0.538 | 0.599 |
| | NE | 0.547 | 0.655 | 51.7 | 61.9 | 0.58 | 0.675 |
| | SE | 0.596 | 0.436 | 34.6 | 25.8 | 0.735 | 0.578 |
| JJA | CONUS | 0.251 | 0.116 | 18.2 | 12.1 | 0.759 | 0.691 |
| | NW | 0.263 | 0.281 | 23.8 | 25.6 | 0.704 | 0.71 |
| | SW | 0.344 | 0.192 | 28.8 | 14.5 | 0.806 | 0.724 |
| | NE | 0.028 | -0.144 | 2.9 | -2.4 | 0.662 | 0.564 |
| | SE | 0.31 | 0.052 | 13.2 | 5.8 | 0.829 | 0.72 |
| SON | CONUS | 0.345 | 0.039 | 48.2 | 9.8 | 0.459 | 0.284 |
| | NW | 0.261 | 0.038 | 56.8 | 9.4 | 0.369 | 0.261 |
| | SW | 0.284 | 0.096 | 55.9 | 20.8 | 0.43 | 0.306 |
| | NE | 0.448 | 0.043 | 47.4 | 5.6 | 0.483 | 0.207 |
| | SE | 0.417 | -0.019 | 32.1 | 2.7 | 0.547 | 0.329 |
| DJF | CONUS | 0.181 | 0.025 | 82.2 | 28 | 0.383 | 0.276 |
| | NW | 0.043 | -0.049 | 77.6 | 40.4 | 0.385 | 0.365 |
| | SW | 0.156 | 0.007 | 70.5 | 19.4 | 0.292 | 0.191 |
| | NE | 0.091 | -0.051 | 96.7 | 14.8 | 0.344 | 0.214 |
| | SE | 0.403 | 0.169 | 87.5 | 33.6 | 0.474 | 0.281 |

Table 3. Similar to Table 3, but based on comparison with FLUXNET-MTE ET during the

period 2000-2011.

| Season | Region | Bias (mm day$^{-1}$) | | Relative bias (%) | | RMSE (mm day$^{-1}$) | |
|--------|--------|------|-------|------|-------|------|-------|
| | | CLM | CLMET | CLM | CLMET | CLM | CLMET |
| Annual | CONUS | 0.207 | 0.065 | 13.3 | 3.2 | 0.328 | 0.24 |
| | NW | 0.07 | 0.013 | 5.8 | 0.0 | 0.222 | 0.234 |
| | SW | 0.051 | -0.047 | 6.8 | -4.7 | 0.244 | 0.241 |
| | NE | 0.309 | 0.165 | 21.9 | 12.2 | 0.334 | 0.238 |
| | SE | 0.427 | 0.154 | 21.3 | 7.6 | 0.461 | 0.248 |
| MAM | CONUS | 0.27 | 0.292 | 15.8 | 19.5 | 0.418 | 0.399 |
| | NW | 0.266 | 0.33 | 22.4 | 28.0 | 0.349 | 0.401 |
| | SW | -0.042 | 0.051 | -7.3 | 2.5 | 0.298 | 0.301 |
| | NE | 0.288 | 0.401 | 21.6 | 30.4 | 0.338 | 0.435 |
| | SE | 0.561 | 0.4 | 26.4 | 18.5 | 0.6 | 0.448 |
| JJA | CONUS | 0.197 | 0.063 | 7.0 | 0.5 | 0.608 | 0.517 |
| | NW | -0.149 | -0.13 | -8.7 | -7.5 | 0.506 | 0.506 |
| | SW | 0.029 | -0.122 | 9.2 | -6.1 | 0.594 | 0.555 |
| | NE | 0.415 | 0.257 | 13.6 | 8.8 | 0.492 | 0.369 |
| | SE | 0.565 | 0.304 | 16.9 | 9.4 | 0.779 | 0.585 |
| SON | CONUS | 0.216 | -0.088 | 20.3 | -9.4 | 0.353 | 0.294 |
| | NW | 0.072 | -0.151 | 9.2 | -22.8 | 0.224 | 0.286 |
| | SW | 0.132 | -0.055 | 21.1 | -5.2 | 0.311 | 0.277 |
| | NE | 0.356 | -0.034 | 33.7 | -1.1 | 0.473 | 0.385 |
| | SE | 0.346 | -0.091 | 21.2 | -5.4 | 0.396 | 0.23 |
| DJF | CONUS | 0.149 | -0.004 | 40.1 | -1 | 0.268 | 0.189 |
| | NW | 0.104 | 0.014 | 27 | -4.9 | 0.279 | 0.26 |
| | SW | 0.086 | -0.063 | 20.9 | -14.4 | 0.17 | 0.129 |
| | NE | 0.176 | 0.037 | 78.5 | 19.2 | 0.329 | 0.208 |
| | SE | 0.236 | 0.002 | 42.8 | 0.8 | 0.282 | 0.129 |

Table 4 Statistics of simulated annual runoff coefficient (ratio of runoff to total precipitation) against GSCD observations over CONUS, Northwest (NW), Southwest (SW), Northeast (NW), and Southeast (SW) during the period 2000-2014.

| | Bias | | Relative bias (%) | | RMSE | |
|---|---|---|---|---|---|---|
| | CLM | CLMET | CLM | CLMET | CLM | CLMET |
| CONUS | -0.053 | -0.027 | -18.5 | -6.7 | 0.198 | 0.192 |
| Northwest | -0.046 | -0.036 | -13.5 | -5.6 | 0.146 | 0.144 |
| Southwest | -0.026 | -0.019 | -19.9 | -11.4 | 0.373 | 0.373 |
| Northeast | -0.06 | -0.022 | -15.7 | -1.5 | 0.108 | 0.092 |
| Southeast | -0.074 | -0.026 | -24.7 | -8.2 | 0.091 | 0.06 |

[Figure]

Figure 1 a) Mean annual (1980-2015) precipitation in mm over conterminous USA (CONUS). NW, SW, NE, and SE represent Northwest, Southwest, Northeast, and Southeast, respectively. The black circles represent sites of in-situ soil moisture observations in Alabama, Illinois, Mississippi, Nebraska, and Oklahoma. b) Locations of the 16 AmeriFlux stations with vegetation types.

[Figure]

Figure 2 Scaling factors of the CLM simulated ET to the GLEAM ET for each month during 1986-1995. The numbers in titles are CONUS-averaged values, and the number of within figures are area-averaged values for each of four sub regions (NW, SW, NE, and SE). The areas with negative scaling factors are masked out.

[Figure]

Figure 3 Mean annual ET from a) GLEAM, b) CLM, and c) CLMET, and the relative difference between d) CLM and GLEAM, e) CLMET and GLEAM, and f) CLMET and CLM during 2000-2014. Numbers in titles are CONUS-averaged values.

[Figure]

Figure 4 Relative bias (RB) for CLM (RB$_{CLM}$), RB for CLMET (RB$_{CLMET}$) during the period 2000-2014, difference in scaling factor f$_{ET}$ between the period 1986-1995 and the period 2000-2014 (f$_{ET}$(86)- f$_{ET}$(00)), and scatter plots of f$_{ET}$(86)- f$_{ET}$(00) versus RB$_{CLMET}$ in January (Jan), April (Apr), July (Jul), and November (Nov).

[Figure]

Figure 5 Mean annual ET from a1) MODIS, b1) FLUXNET-MTE, and the relative differences between a2) CLM and MODIS, b2) CLM and FLUXNET-MTE, a3) CLMET and MODIS, and b3) CLMET and FLUXNET-MTE during 2000-2011. Numbers in titles are CONUS-averaged values.

[Figure]

Figure 6 Seasonal cycles of ET from MODIS, FLUXNET-MTE, CLM, and CLMET over CONUS, Northwest, Southwest, Northeast, and Southeast during the period 2000-2011.

[Figure]

Figure 7 Time series of ET difference between model (CLM or CLMET) and reference data (MODIS or FLUXNET-MTE) over CONUS, Northwest, Southwest, Northeast, and Southeast during the period 2000-2011.

[Figure]

Figure 8 Monthly mean latent heat fluxes from CLM, CLMET and observations at 16 flux tower sites. RMSECLM and RMSECLMET represent the root mean square error against observations for CLM and CLMET, respectively. Note that the CLM and CLMET simulations are driven with meteorological forcing at the grid cell level (as opposed to site-specific forcing).

[Figure]

Figure 9 Daily mean latent heat fluxes from CLM and CLMET grids and station observations at

ARM SGP Burn, Audubon Grassland, Bondville, Donaldson, Flagstaff Forest, Fort Dix, Fort

Peck, and Little Prospect. RMSECLM and RMSECLMET represent the root mean square error

against observations for CLM and CLMET, respectively.

[Figure]

Figure 10 Daily mean latent heat fluxes from CLM and CLMET grids and station observations at Mead Rainfed, Metolius Pine, Missouri Ozark, Morgan Forest, Sylvania Wilderness, Tonzi Ranch, Walnut River, and Wind River Crane. RMSECLM and RMSECLMET represent the root mean square error against observations for CLM and CLMET, respectively.

[Figure]

Figure 11 Mean annual runoff coefficient (the ratio runoff to total precipitation) from a) Global Streamflow Characteristics Dataset (GSCD), b) CLM, and c) CLMET, and the relative differences between d) CLM and GSCD, e) CLMET and GSCD, and f) CLMET and CLM during 2000-2014. Runoff coefficient less than 0.02 is blanked out. Numbers in titles are CONUS-averaged values.

[Figure]

Figure 12 Surface runoff and subsurface runoff simulated in CLM and CLMET and their relative differences during 2000-2014. Numbers in titles are the CONUS-averaged values.

[Figure]

Figure 13 Simulated soil moisture (mm) in the top 0-10 cm and 0-100 layers in August from CLM and CLMET, their differences, and their relative differences during 2000-2014.

[Figure]

Figure 14 Monthly volumetric soil water content ($m^{-3}$ $m^{-3}$) in the top 0-10cm soil layer from the quality-controlled NASMD, CLM, and CLMET over the state of Alabama (AL), Illinois (IL), Mississippi (MS), Nebraska (NE), and Oklahoma (OK) for the period of 2006-2010.

[Figure]

Figure 15 Same as Figure 14, but for the top 0-100cm soil layer.

---

## Referee Report (RR1)

Wang *et al.* (2017): Incorporating remote sensing ET into Community Land Model version 4.5

GENERAL COMMENTS

In general, the authors implemented most of my initial suggestions and replied properly to the comments raised by the two reviewers. Below, I list some final comments.

SPECIFIC COMMENTS

1. At P12-Sect. 4.1, the authors now clearly mention that no scaling is applied during periods where the sign of evaporation does not agree between the estimate from GLEAM, and the original CLM run. As can be deduced from Figure 2, this mainly occurs during winter months. I think it should be clearly mentioned in the paper what the impact of this decision is/may be. My guess is that this will introduce a seasonal bias in the "corrected" evaporation, or will even result in discontinuities (especially during the transition of a month with scaling factor to a month without scaling factor).

2. It is obvious from the results in Figures 3 and 5, that the impact of the bias correction algorithm is very limited in the western part of the CONUS. This is also acknowledged several times in the paper, but no clear reason is given why this is the case. As CLM tends to underestimate the GLEAM-derived evaporation in some parts of the West (mainly near the coast), and evaporation in these regions is mainly limited by water, I guess the low impact there could be related to the reasons mentioned at the end of P15. However, the reason explained along these lines is not valid in case of an initial overestimation of CLM, which is for instance happening in the eastern part of the North-West region (see Figure 3d). Also in that region, the impact of the bias correction algorithm is very low and I think the reason for this should be figured out and explained in the text.

3. Related to the previous comment, I do not agree with the statement at P14: "*Note that there is still a substantial overestimation in western CONUS in CLMET compared with the MODIS ET, partially because the algorithm developed by Mu et al. (2007, 2011) underestimate ET in the MODIS product (Michel et al. 2016, Miralles et al. 2016).*" I fully agree with the fact that the MODIS algorithm generally produces lower estimates of evaporation, probably explaining the severe positive bias that can be seen in Figure 5a2. However, the reason why this bias in the West is not reduced in CLMET is because the bias correction algorithm has simply no impact in that region (see also the previous comment). As a result, the bias is still present in CLMET. In addition, the negative bias between CLM and the FLUXNET-MTE product in the West is also not alleviated, while this is not discussed in the manuscript.

4. I think the fact that the method can hardly deal with underestimations of evaporation in water-limited regimes (see discussion at the end of P15) is an important drawback of the method and somehow summarizes the oddness of technique. This shows that this is a pure post-processing method and that it is not able to fix the real problem, which lies somewhere in the model physics. Therefore, I think that this should be highlighted in the conclusion as well.

**TECHNICAL CORRECTIONS**

1. Please add a reference to the numbers of global evaporation and runoff listed near the end of P3.

2. The paper should be carefully checked for typos and the proper use of English grammar. I am only listing here a few examples, but the list is not limited to the ones below:
   a. P6: "We **follows → follow**"
   b. P6: "Although land surface models are **cable → capable**"
   c. P13: "All the statistics in CLMET **is → are**"
   d. P15: "**CNOUS → CONUS**"
   e. P15: "... in contrast, when an **underestimate → underestimated**"
   f. P17: "... scaling factor within each **gird → grid**"
   g. P20: "... in this study can effectively **improves → improve**"

3. The paper has currently 15 figures and 4 tables (which could be summarized in figures as well to strengthen the message, as the tables are not really giving a nice overview). I do not know the regulations of Copernicus Publications, but 15 figures seems a lot. Maybe, the authors should think about reducing the number of figures and only keeping the key figures.

*Brecht Martens*
*Ghent University*
*Laboratory of Hydrology and Water Management*

---

## Author Response (AR2)

**Responses to the comments from Reviewer #1**

HESS-2016-696
Wang et al. (2017): Incorporating remote sensing ET into Community Land Model version 4.5

GENERAL COMMENTS

In general, the authors implemented most of my initial suggestions and replied properly to the comments raised by the two reviewers. Below, I list some final comments.

SPECIFIC COMMENTS

1.At P12-Sect. 4.1, the authors now clearly mention that no scaling is applied during periods where the sign of evaporation does not agree between the estimate from GLEAM, and the original CLM run. As can be deduced from Figure 2, this mainly occurs during winter months. I think it should be clearly mentioned in the paper what the impact of this decision is/may be. My guess is that this will introduce a seasonal bias in the "corrected" evaporation, or will even result in discontinuities (especially during the transition of a month with scaling factor to a month without scaling factor). Response: Point well taken. For example, in the eastern part of the North-West region, as pointed out in comment 2 by the reviewer, because no scaling factors are applied in November and December, the ET bias for CLMET are still substantial during the winter months (which partially explains the lower bias effect on the annual simulated ET, as shown in Figures 3e-3g). In contrast, the ET bias during the summer months is substantially corrected by applying the scaling factors. We have added a sentence to point out the potential effect of this treatment.

"*Figure 2 shows the climatological scaling factors for each month over CONUS based on the 1986-1995 period. The GLEAM-derived dew and the CLM simulated dew is not consistent in some areas of northwest CONUS. If that happens, the scaling factors became negative, because ET is negative for one and positive for the other. We did not scale ET when the scaling factor is negative, and those areas are masked out in Figure 2. **This treatment (scaling in some months and no scaling in other months) may introduce a seasonal bias correction effect in these areas…**" (page 12, lines 250-251)

2. It is obvious from the results in Figures 3 and 5, that the impact of the bias correction algorithm is very limited in the western part of the CONUS. This is also acknowledged several times in the paper, but no clear reason is given why this is the case. As CLM tends to underestimate the GLEAM-derived evaporation in some parts of the West (mainly near the coast), and evaporation in these regions is mainly limited by water, I guess the low impact there could be related to the reasons mentioned at the end of P15. However, the reason explained along these lines is not valid in case of an initial overestimation of CLM, which is for instance happening in the eastern part of the

North-West region (see Figure 3d). Also in that region, the impact of the bias correction algorithm is very low and I think the reason for this should be figured out and explained in the text.

Response: Several factors influence effect of the bias correction algorithm: the difference in the scaling factor between the calibration period and the validation period (as shown in Figure 4 and discussed in page 14), whether ET is overestimated or underestimated (as shown in Figure 7 and discussed in page 15), and whether scaling is applied or not (as shown in Figure 2 and discussed in page 12). The latter two factors are the primary reasons for the smaller improvement in the eastern part of the North-West region. We have added a paragraph in Section 5, together with a previously existing paragraph in the revised format, to explain why the bias correction method performs differently for different areas.

*"Qualitatively, whether the Parr et al. (2015) ET bias correction approach improves the quantification of the hydrological cycle depends on whether ET is limited by water or energy and whether ET is underestimated or overestimated. The approach works well when/where ET is not limited by water availability; in water-limited regimes, the approach is effective in correcting the positive ET biases but does not work well if ET is underestimated. Quantitatively, the degree of the model improvement derived from this bias correction algorithm is highly related to whether the fundamental assumption of Parr et al. (2015) (on a time-invariant relationship characterizing the default model biases) holds or not. Although the scaling factors between observations and simulations do not change much from the calibration period to the validation period over most regions in most seasons, dramatic changes do exist in some areas. Differences in the scaling factors between the calibration and verification/application periods greatly influence the effectiveness of the bias correction method, with large differences causing the approach to be less effective leaving substantial biases in CLMET. Northeast CONUS during winter is an example of having a large bias in CLMET due to greater changes in the ET scaling factor from the calibration period to the verification period.*

*Another factor affecting the degree of the model improvement is whether the ET scaling is applied at all. As shown in Figure 2, we do not scale ET in some areas of Northwest CONUS during the winter months due to the inconsistence in the ET sign (positive or negative) between GLEAM and CLM. In these areas and season(s), ET in CLMET is not corrected at all. All these three factors (i.e., whether the scaling factor differs significantly between calibration and validation periods, whether ET is underestimated in water-limited regimes, and whether ET scaling is applied at all) influence the effectiveness of the bias correction approach, but one or two of them may dominate for a given region/season. **For example, regardless of which product is used as the reference for comparison (Figures 3g, 5a4, 5b4), the approach reduces ET biases over the eastern CONUS where the ET scaling is applied in most places/seasons and the scaling factor shows little difference between the calibration and validation periods. In contrast, in the north part of the Midwest, some positive biases still remain in CLMET, as the ET scaling is not applied in winter months and***

*the scaling factor differs quite much between these two periods. Over a portion of western CONUS, the bias correction approach is less effective due to the underestimation of ET under a water-limited condition and large differences between calibration and validation periods in the scaling factor."* (page 19, lines 413-443)

3. Related to the previous comment, I do not agree with the statement at P14: "Note that there is still a substantial overestimation in western CONUS in CLMET compared with the MODIS ET. partially because the algorithm developed by Mu et al. (2007, 2071) underestimate ET in the MODIS product (Michel et al. 2016, Miralles et al. 2016)." I fully agree with the fact that the MODIS algorithm generally produces lower estimates of evaporation, probably explaining the severe positive bias that can be seen in Figure 5a2. However, the reason why this bias in the West is not reduced in CLMET is because the bias correction algorithm has simply no impact in that region (see also the previous comment). As a result, the bias is still present in CLMET. In addition, the negative bias between CLM and the FLUXNET-MTE product in the West is also not alleviated, while this is not discussed in the manuscript.

Response: We have deleted the sentence "partially because the algorithm developed by Mu et al. (2007, 2071) underestimate ET in the MODIS product (Michel et al. 2016, Miralles et al. 2016).", and added a paragraph in Section 5 to explain why the bias correction method performs differently for different areas (including west CONUS). Please see the response to comment 2.

4. I think the fact that the method can hardly deal with underestimations of evaporation in water-limited regimes (see discussion at the end of Pl5) is an important drawback of the method and somehow summarizes the oddness of technique. This shows that this is a pure post-processing method and that it is not able to fix the real problem, which lies somewhere in the model physics. Therefore, I think that this should be highlighted in the conclusion as well.

Response: We have added a sentence on the drawback of the method in Section 5.

*"Qualitatively, whether the Parr et al. (2015) ET bias correction approach improves the quantification of the hydrological cycle depends on whether ET is limited by water or energy and whether ET is underestimated or overestimated. **The approach works well when/where ET is not limited by water availability; in water-limited regimes, the approach is effective in correcting the positive ET biases but does not work well if ET is underestimated** ....."*(page 19, lines 413-417)

TECHNICAL CORRECTIONS

1. Please add a reference to the numbers of global evaporation and runoff listed near the end of P3.

Response: We have added a reference for this in the revised manuscript.

*"Haddeland et al. (2011) compared eleven models in simulating evapotranspiration*

*(ET), and found that the global ET on land surface ranges from 415 to 586 mm year-1 and the runoff ranges from 290 to 457 mm year-1.*" (page 3, lines 63-66)

2. The paper should be carefully checked for typos and the proper use of English grammar. I am only listing here a few examples, but the list is not limited to the ones below:
Response: We have thoroughly checked manuscript, and made corrections accordingly, including correcting the errors listed below.

a.  P6: "We follows → follow"
b.  P6: "Although land surface models are cable → capable"
c.  P13: "Allthe statistics in CLMET is → are"
d.  P15: "CNOUS → CONUS"
e.  P15: "... in contrast, when an underestimate→ underestimated"
f.  P17: "... scaling factor within each gird→ grid"
g.  P20: "... in this study can effectively improves → improve"

3.  The paper has currently 15 figures and 4 tables [which could be summarized in figures as well to strengthen the message, as the tables are not really giving a nice overview). I do not know the regulations of Copernicus Publications, but 15 figures seems a lot. Maybe, the authors should think about reducing the number of figures and only keeping the key figures.
Response: We have deleted previous Figures 12, 14, and 15, which results in 12 figures in total in the revised manuscript.

**Responses to the comments from Reviewer #2**

General comments

Although I was not part of the first review round, I generally agree with the comments from the previous reviewers. That is, the paper describes a simple approach to correct for biases in CLM simulations of evaporation over CONUS by rescaling the model ET to GLEAM. Although the approach is not really innovative and does not result in improved physical understanding, it is effective in reducing biases and improving state and flux estimates, and therefore, can be of interest to the community. I also believe the authors have substantially improved the manuscript after revision, particularly by including additional independent evaporation datasets (other than GLEAM) for validation, also at the daily time scale. Below, I still formulated some minor comments, which I believe could help further improving the manuscript. For future reference, please include line numbers in the manuscript to help tracking comments.
Response: Thank the reviewer for the positive evaluation. We have included line numbers in the revised manuscript.

Specific comments

1. Title: given that GLEAM evaporation is more of a model product (forced by remote sensing observations), than an actual remote sensing product, I would recommend to change the title accordingly.

Response: Following the reviewer's suggestion, we have changed the tile to "Incorporating remote sensing-based ET estimates into Community Land Model version 4.5".

2. Introduction/methods: I think it is important to add a caveat somewhere in the introduction or method section, in that the approach assumes full trust in the GLEAM evaporation, and no trust at all in the CLM evaporation. From a data merging/assimilation perspective, it could be more desirable if information from both CLM and GLEAM would be exploited, based on their relative uncertainties/performances. In its current form, the approach can degrade the performance of ET estimates in places/periods where GLEAM evaporation may perform less well. For instance, GLEAM has a very simple snow module, so it might have issues over snow-covered areas, like the Sierra Nevada (an area that actually stands out in Figure 2).

Response: We have added a sentence to show full trust in the GLEAM evaporation in Section 3 when the GLEAM ET is first described in the manuscript.

"*As such we assume full trust in the GLEAM evaporation data with the bias correction method.*" (Page 9, lines 179-180)

3. Page 4, 3rd paragraph: The authors mention data assimilation as another approach for reducing biases. I would rephrase this part, as the general purpose of data assimilation is not to remove biases, but to correct for random errors (e.g. in the forcings). Biases between model forecasts and observations should be corrected prior to assimilation.

Response: we have changed from "reducing model biases" to "improving model simulations or predictions"

"*Another approach to **improving model simulations or predictions** is through data assimilation, by merging observational data and land surface models to obtain optimal estimates for next time step*" (page 4, line 82)

4. Page 6, 1st paragraph: The authors state that CLM can realistically capture the spatial pattern of ET over CONUS. This seems to be contradictory to what is shown later in the results section, where large biases occur over space (e.g. Figure 2) and time. Maybe this statement needs to be nuanced, by mentioning the scale to which it applies, or mentioning the reference dataset on which it is based.

Response: we have changed to "overall spatial pattern of ET".

"*Parr et al. (2016) found that CLM4.5 can realistically capture the **overall** spatial*

*pattern of ET in CONUS when the model is forced by the NLDAS-2 meteorological variables*" (page 5, lines 115-116)

5. Page 6, 3rd paragraph: I was wondering if, by applying constant monthly scaling factors, there may be jumps in simulated states/fluxes from one month to the other. Alternatively, the authors could consider interpolating scaling factors to get a smooth transition. Please comment on whether such jumps are observed in the simulations (e.g. of daily time scale runoff, soil moisture, etc.)

Response: The scaling factor characterizes the relationship between model biases and ET climatology, and the fundamental assumption is that the nature of the model biases is time-invariant at the inter-annual and longer time scales. The monthly time scale is used here to account for its seasonality. To say that the nature of the model biases varies on a finer temporal scale (e.g., day-to-day time scale) does not make physical sense, although technically it can be done. In fact we tested the performance of CLMET based on daily scaling factors. CLMET performance is not improved using daily scaling factors as compared with CLMET using monthly scaling factors.

6. Page 7, 2nd paragraph: Please describe which adjustments are made in CLM if soil moisture cannot support the evaporative demand. This is a quite important aspect in view of the approach.

Response: We have specified the adjustments as explained in the following paragraph:

*"Since the overwriting process in CLMET may break the water balance, the model checks whether the amount of water stored in vegetation canopy is sufficient to sustain the interception loss and whether the surface soil water storage is sufficient to sustain soil evaporation through the model time step. If not, the interception loss (soil evaporation) rate is set to be equal to the water available in vegetation canopy (soil) divided by the model time step. This adjustment minimizes the imbalance caused by overwriting ET components in CLMET."* (page 7, lines 153-159)

7. Equations 1-5: Are absolute values of S and R considered for calculating statistics like bias? If not, and when averaging over grid cells, simulations which have negative bias in one place and positive bias in another place would show up as bias-free, which is incorrect.

Response: It is true that the positive and negative biases are canceled out when they are averaged over grid cells. However, the region-averaged bias can provides us with the systematic error of land surface models, and this statistic has been widely used in model evaluation studies. In fact, we tested evaluating the model performance based on the absolute bias, and the conclusion is consistent with the bias-based conclusion, e.g., larger improvement from CLMET to CLM in eastern CONUS, larger improvement during autumn and winter.

That said, to address the reviewers concern, we have added plots for spatially distributed changes in the absolute value of relative biases (Figures 3g, 5a4, 5b4) to provide a more clear picture of the performance improvement from CLM to CLMET.

8. Results: I believe this section would improve if the authors would at least try to formulate some hypotheses on why biases in CLM simulations of ET occur over some areas and time periods. This could particularly help future studies to improve physical model aspects. Based on your extensive validation analysis of CLM ET, what can be learned? Are ET biases likely originating from biases in soil moisture, biases in radiation/temperature, model physics or parameters (and if so, which parameters)? Is this different for different regions, e.g. water- vs energy-limited regions, etc.?

Responses: We have added one paragraph to specifically talk about the physical conditions (water- or energy-limited ET regimes) that influence the effectiveness of the approach, which serves as the basis for another short paragraph at the end of paper on the implications of our results for physically-based land surface model development:

"...... *Figure 7 compares the temporal evolution of the simulated ET in CLM and CLMET against MODIS and FLUXNET-MTE ET over CONUS and four sub-regions. It is evident that the bias correction method in CLMET is very effective in reducing overestimation (positive bias), but does not work as well in correcting the underestimation (negative bias) in water-limited regimes. The difference has to do with the specific ET regime, i.e. whether ET is limited by water or energy. When an overestimated ET is overwritten with a lower value, the water on land is sufficient to support the reduced ET; in contrast, when an underestimated ET is overwritten with a higher value, the land surface model checks whether water storage in soil layer and vegetation canopy can sustain the elevated ET and further adjust if necessary to keep with the mass conservation equation. The extent to which ET can be increased is limited by the availability of water stored in soil layer and vegetation canopy. Therefore, in water-limited ET regimes, if ET is underestimated in CLM, the actual ET in CLMET after the water availability check can be substantially lower than the corrected ET fed into the model, which diminishes the effect of the bias correction algorithm under such circumstance.*" (page 15, lines 327-340)

"......*However, results from this study can provide useful guidance for physically-based land surface model development. As can be seen from Figure 3g, the bias correction algorithm improves ET estimation over most of the U.S., indicating a strong potential for performance improvement that can be derived from improving the physical parameterization of ET processes in the model. Over regions where the bias correction approach does not improve the ET estimate (which are mostly places where ET is water-limited while the model underestimates ET), parameterizations for other processes that influence soil moisture (e.g., runoff generation, groundwater interactions) are the most likely cause for model biases and should be the focus of physically-based model development effort.*" (page 22, lines 472-480)

9. Page 13, 2nd paragraph, line 12: I believe it is important to stress that improvements in model performance are relative to GLEAM. GLEAM has its own shortcomings, so improving towards GLEAM not necessarily means improving estimates overall.

Response: We have pointed out that the evaluation in the model performance are relative to the GLEAM ET at the beginning of this paragraph in the revised manuscript.

"*Over most of CONUS, CLM overestimates ET and CLMET reduces ET as well as ET biases relative to GLEAM data.*" (page 13, lines 274-276)

10. Page 20, 2nd paragraph: I appreciated the statement that the approach does not replace model improvements through better parameterization of physical processes. This frames the value of your approach, and helps readers to identify shortcomings and advantages of the approach.
Response: We thank the reviewer for the positive evaluation.

Technical corrections

I would strongly encourage the authors to perform an in-depth check of the grammar throughout the manuscript. There are still many mistakes, e.g., on using plural nouns and tenses. The following list is not comprehensive.
Response: We have thoroughly checked manuscript, and made corrections accordingly, including (but not limited to) a long list of corrections suggested by this reviewer.

Figures: Please make sure you refer to panels (a, b, c, …) in all figure captions (for instance figure 4)
Response: Following reviewer's suggestion, we have changed the presentation style in all figure captions.

**Incorporating remote sensing-based ET estimates into Community Land**

**Model version 4.5**

Dagang Wang[1, 2, 3, 4]\*, Guiling Wang[4]\*, Dana T. Parr[4], Weilin Liao[1, 2], Youlong Xia[5], Congsheng

Fu[4]

[1] School of Geography and Planning, Sun Yat-sen University, Guangzhou, China

[2] Guangdong Key Laboratory for Urbanization and Geo-simulation, Sun Yat-sen University,

Guangzhou, China

[3] Key Laboratory of Water Cycle and Water Security in Southern China of Guangdong High

Education Institute, Sun Yat-sen University, Guangzhou, P.R. China

[4] Department of Civil and Environmental Engineering, University of Connecticut, Storrs, USA

[5] National Centers for Environmental Prediction/Environmental Modeling Center, and I. M.

System Group at NCEP/EMC, College Park, Maryland, USA

Submitted to special issue of Hydrology and Earth System Sciences: Observations and modeling of land surface water and energy exchanges across scales, in Honor of Eric F. Wood

Revised June, 2017

\*Corresponding authors: Dr. Dagang Wang, School of Geography Science and Planning, Sun

Yat-sen University, Guangzhou, P. R. China 510275, [wangdag@mail.sysu.edu.cn](mailto:wangdag@mail.sysu.edu.cn), (86)

2084114575. Dr. Guiling Wang, Department of Civil and Environmental Engineering,

University of Connecticut, Storrs, USA, [guiling.wang@uconn.edu](mailto:guiling.wang@uconn.edu)

**Abstract**

Land surface models bear substantial biases in simulating surface water and energy budgets despite the continuous development and improvement of model parameterizations. To reduce model biases, Parr et al. (2015) proposed a method incorporating satellite-based evapotranspiration (ET) products into land surface models. Here we apply this bias correction method to the

Community Land Model version 4.5 (CLM4.5) and test its performance over the conterminous US

(CONUS). We first calibrate a relationship between the observational ET from the Global Land

Evaporation Amsterdam Model (GLEAM) product and the model ET from CLM4.5, and assume that this relationship holds beyond the calibration period. During the validation or application period, a simulation using the default CLM4.5 ("CLM") is conducted first, and its output is combined with the calibrated observational-vs-model ET relationship to derive a corrected ET; an experiment ("CLMET") is then conducted in which the model-generated ET is overwritten with the corrected ET. Using the observations of ET, runoff, and soil moisture content as benchmarks, we demonstrate that CLMET greatly improves the hydrological simulations over most of CONUS, and the improvement is stronger in the eastern CONUS than the west and is the strongest over the southeast CONUS. For any specific region, the degree of the improvement depends on whether the relationship between observational and model ET remains time-invariant (a fundamental hypothesis of the Parr et al. method) and whether water is the limiting factor in places where ET

is underestimated. While the bias correction method improves hydrological estimates without improving the physical parameterization of land surface models, results from this study does provide guidance for physically based model development effort.

[revised manuscript text omitted]
 whether the amount of water stored in vegetation canopy is sufficient to sustain the interception loss and whether the surface soil water storage is sufficient to sustain soil evaporation through the model time step. If not, the interception loss (soil evaporation) rate is set to be equal to the water available in vegetation canopy (soil) divided by the model time step. This adjustment minimizes the imbalance caused by overwriting ET components in CLMET.

In this study, the statistics Bias, Relative bias, and root mean square error (RMSE) are used to validate models in reproducing the spatial pattern against the reference dataset. They are defined as:

$$Bias = \frac{1}{N} \sum_{i=1}^{i=N} \left( \overline{S_i} - \overline{R_i} \right) \tag{1}$$

$$Relative\ bias = \frac{1}{N} \sum_{i=1}^{i=N} \frac{\left( \overline{S_i} - \overline{R_i} \right)}{\overline{R_i}} \tag{2}$$

$$RMSE = \sqrt{\frac{\sum_{i=1}^{i=N} \left( \overline{S_i} - \overline{R_i} \right)^2}{N}} \tag{3}$$

Where N is the total number of grid cells, and $\overline{S_i}$ ( $\overline{R_i}$ ) are the temporal average of model simulated (reference) value for grid cell i, which is calculated as:

$$\overline{S_i} = \frac{1}{M} \sum_{j=1}^{j=M} S_{i,j} \tag{4}$$

$$\overline{R_i} = \frac{1}{M} \sum_{j=1}^{j=M} R_{i,j} \tag{5}$$

Where $S_{i,j}$ ($R_{i,j}$) is model simulated (reference) value at time j and grid cell i, M is the total number of time points. The statistic RMSE is also used to validate models in reproducing time series where M becomes the total number of grid cells and N the total number of time points.

**3 Data**

3.1 ET

3.1.1 GLEAM ET

GLEAM (The Global Land Evaporation Amsterdam Model) version 3.0a (Miralles et al.

2011, Martens et al. 2016) is used to calibrate the ET scaling factors and to validate CLM and

CLMET. As such we assume full trust in the GLEAM evaporation data with the bias correction method. GLEAM 3.0a was derived based on reanalysis net radiation and air temperature, a combination of gauge-based, reanalysis and satellite-based precipitation and satellite-based vegetation optical depth, spanning the 35-year period 1980–2014 (http://www.gleam.eu/).

Potential evaporation in GLEAM 3.0a was calculated using a Priestley and Taylor equation based on surface net radiation and near-surface air temperature, and was converted to actual evaporation using the multiplicative evaporative stress factor. The dataset has been used in studying soil moisture-temperature coupling (Miralles et al. 2012), the impact of land surface on precipitation (Guillod et al. 2015), and the climate control on land surface evaporation (Miralles et al., 2014).

Recent evaluations conducted at both flux tower site and global scales show that GLEAM-based

ET is superior to MODIS-based and the Surface Energy Balance System (SEBS) based ET

products (Michel et al. 2016, Miralles et al. 2016). The spatial resolution of GLEAM dataset is

0.25°, which is consistent with the resolution of CLM4.5 used in this study. The temporal resolution of GLEAM dataset is daily, and the monthly aggregated ET is used to derive the scaling factors.

3.1.2 MODIS and FLUXNET-MTE ET

Two other gridded ET products are used for independent evaluations: MODIS ET and

FLUXNET-MTE (model tree ensemble) ET. Mu et al. (2007, 2011) produced a MODIS-based global ET dataset using a revised Penman–Monteith (PM) equation. The dataset is arguably the most widely used remote sensing-based global ET product (Miralles et al. 2016). Monthly version of the MODIS-based product at the 0.5° spatial resolution are used to validate the model with the bias correction method. The FLUXNET-MTE global ET dataset was derived from 253 FLUXNET

eddy covariance towers distributed over the globe using the model tree ensemble (MTE) approach (Jung et al., 2009, 2010). The record gaps of half hourly eddy covariance fluxes were filled first, and the complete tower-based dataset was then used to train MTE to produce monthly global ET

dataset at the 0.5° spatial resolution. The data have been used to study the ET trend (Jung et al.,

2010) and to improve canopy processes in a land surface model (Bonan et al., 2011). As

FLUXNET sites over the CONUS are fairly dense, the quality of the FLUXNET-MTE dataset in our study domain is expected to be good. The MODIS dataset is available for 2000-2014, and the

FLUXNET-MTE dataset is available for 1982-2011. We chose the overlap period of these two products, 2000-2011, for model validations using MODIS and FLUXNET-MTE dataset.

3.1.3 Flux Tower ET

ET observations (in energy unit) at 16 sites from the AmeriFlux network are used to validate the model on the grid cell scale (Figure 1b). Those sites span four sub-regions (i.e., NW,

SW, NE, and SW) of CONUS with five different vegetation types (i.e., grass, crop, evergreen needleleaf forest, mixed forest, and deciduous broadleaf forest). More details about these flux tower sites can be found in Xia et al. (2015b). For most sites, the year of 2005 is selected for validation because data for this year has the least amount of missing records; three sites are exceptions due to data availability: 2002 for the site of Sylvania Wilderness, 2004 for the sites of

Donaldson and Walnut River. Both daily and monthly ET observations at these 16 sites are compared with model simulations.

3.2 Observation-based Runoff Coefficient

The runoff coefficient (the ratio of runoff to precipitation) of Global Streamflow

Characteristics Dataset (GSCD) version 1.9 (Beck et al., 2013, Beck et al., 2015) is used to evaluate the model performance in simulating runoff. The GSCD dataset was produced based on streamflow observations from approximately 7500 catchments over the globe. A data-driven approach was adopted to derive the gridded streamflow characteristics at the 0.125° resolution on a global scale. This dataset is relatively reliable for the grid cells within which a large number of catchments data is used. The uncertainty is low in North America, Europe, and southeastern

Australia where a large number of observations are available.

3.3 In-situ soil moisture observations

The North American Soil Moisture Database (NASMD) is used to evaluate the model performance in simulating soil moisture in both the surface (0-10cm) and root-zone (0-100cm)

layers. NASMD was initiated in 2011 to provide support for developing climate forecasting tools, calibrating land surface models, and validating satellite-derived soil moisture algorithms. A

homogenized procedure has been implemented, as the measurement stations are across a variety of in-situ networks. In addition, a quality control (QC) algorithm was applied to the measurement records (Xia et al., 2015; Liao et al., submitted to Journal of Hydrometeorology, 2017). The in- situ observations in Alabama (AL), Illinois (IL), Mississippi (MS), Nebraska (NE), and Oklahoma (OK) from 2006-2010 are selected from NASMD (Figure 1a). A large number of stations is evenly distributed over these states and observation records during this period are relatively complete after QC. The numbers of stations in AL, IL, MS, NE, and OK are 10, 19, 14, 45, 105, and 39, respectively. Since the soil layer where measurement was taken varies with stations, we linearly interpolate the volumetric soil water content to the 5 cm and 50 cm depth for all stations to compare with the modeled soil moisture for the 0-10 cm and 0-100 cm layers.

**4 Results**

4.1 Calibration of ET Scaling Factor

Figure 2 shows the climatological scaling factors for each month over CONUS based on the 1986-1995 period. The GLEAM-derived dew and the CLM simulated dew is not consistent in some areas of northwest CONUS. If that happens, the scaling factors became negative, because

ET is negative for one and positive for the other. We did not scale ET when the scaling factor is negative, and those areas are masked out in Figure 2. ==This treatment (scaling in some months and==

==no scaling in other months) may introduce a seasonal bias correction effect in these areas.== The model simulations generally agree better with GLEAM estimations during the warm seasons, whereas the difference between simulations and GLEAM estimations remains large during the cold seasons. The scaling factors greatly vary with region. For instance, the area-averaged scaling factors for November are 0.34, 0.58, 0.28, and 0.52 for Northwest, Southwest, Northeast, and

Southeast, respectively. The overestimation is overwhelming during October, November,

December, and January, whereas underestimation occurs in many areas during March, April, and

May. The overestimation is especially severe over the Northeast CONUS where simulated ET is almost five times of GLEAM estimate in December.

4.2 Evaluation

We evaluate the effectiveness of the ET bias correction approach in CLM4.5 by comparing results from CLM and CLMET with the reference dataset. The evaluation metrics examined include bias, relative bias, and root mean square error (RMSE) as described in Section 2.2. Since the spatial resolution of some gridded reference data is not consistent with the model resolution, we upscale the finer resolution data to match the coarser resolution data using simple arithmetic averages.  For example, when the MODIS and FLUXNET-MTE ET are used for validation, we average ET from the four 0.25° model grid cell within each 0.5° observational grid cell; for the GSCD runoff data, we aggregate observations from 0.125° to 0.25° to match the model resolution. As in-situ soil moisture observations are technically at the point scale, we spatially average observed soil moisture in each state and compare the averaged observations with the model simulations averaged across grid cells within the same state.

4.2.1 ET

Figure 3 shows the multi-year averages (2000-2014) of ET derived from GLEAM, simulated by CLM and CLMET, and the relative bias of simulations against GLEAM. Over most of CONUS, CLM overestimates ET and CLMET reduces ET as well as ET biases relative to

GLEAM data. The averaged relative bias in CLM over CONUS is 10.8%, with relative bias exceeding 10% in a substantial portion of CONUS; and in CLMET, the CONUS-averaged relative bias is reduced to -0.1%, and it is within 10% over most of CONUS. This improvement is more significant over eastern CONUS than western CONUS. Table 1 shows the statistics on the model performance with these two schemes during different seasons and in four sub regions. CLM

overestimates the CONUS-averaged ET in all other seasons except for March-April-May (MAM), and the largest overestimation occurs in Northeast CONUS during December-January-February (DJF) with a relative bias as large as 146.4%. The underestimation in MAM is largest over

Southwest CONUS with a relative bias of -17.9%. CLMET substantially improves the model performance as indicated by the various metrics. All the statistics in CLMET are superior to those in CLM with a few exceptions in bias or relative bias. The improvement from CLM to CLMET is more substantial for September-October-November (SON) and DJF than MAM and June-July-

August (JJA). The relative bias of 51% (77.7%) in CLM is reduced to 7.8% (18.9%) in CLMET

over CONUS during SON (DJF). For the regional average, the improvement is greatest over

Southeast CONUS. All the positive biases in all seasons over Southeast CONUS are substantially reduced.

To understand the differences between CLM and CLMET, we select four months representing each of the four seasons, January, April, July, and November, to examine the relationship between the relative bias of model simulations and the scaling factor changes from calibration period (1986-1995) to validation period (2000-2014) in Figure 4. The improvement from CLM to CLMET is evident, especially in January and November (Figure 4a-4b). Although the bias is dramatically reduced in CLMET, it remains large in Northeast CONUS in January (Figure 4b1). In addition, the bias in CLMET appears larger in western CONUS than eastern

CONUS (Figure 4b). The spatial patterns of the relative biases in CLMET and the scaling factor differences between the two periods demonstrate a great degree of similarity (Figure 4b-4c), and the scatter plots between the two quantities (Figure 4d) reflect a strong correlation. Not surprisingly, the degree to which CLMET can improve model performance in simulating ET greatly depends on how stable the scaling factor is from the calibration period to the validation period, i.e., how well the assumption of a time-invariant scaling relationship holds. Over most of CONUS, changes in the scaling factor are within 10% (Figure 4d). This temporal stability of the relationship between observed ET and simulations guarantees improvements from CLM to CLMET.

CLM and CLMET performances are also evaluated using two independent observation datasets of ET, MODIS-based and FLUXNET-MTE-based ET (Figure 5, Tables 2 and 3). For the multi-year averaged ET, the relative bias in CLMET is smaller than that in CLM, and the improvement is greater in eastern CONUS than western CONUS as compared with either MODIS- or FLUXNET-MTE-based ET. Note that there is still a substantial overestimation in western

CONUS in CLMET compared with the MODIS ET. With the reference of the MODIS or

FLUXNET-MTE ET, CLMET corrects bias for all other three seasons except for MAM (Tables 2

and 3). Bias, relative bias and RMSE in CLMET are greater than in CLM for the whole CONUS,

Northwest, Southwest, and Northeast in MAM. The most considerable improvement occurs in

SON compared with the other three seasons. CLMET deteriorates the ET estimate for MAM by enhancing the overestimation already occurring in CLM, which is different from the validation against the GLEAM-based ET.

The analysis on time series of ET from MODIS, FLUXNET-MTE, and the two types of simulations also demonstrates improvement from CLM to CLMET. Climatological seasonal cycles of ET over CONUS and four sub regions for the period 2000-2011 are shown in Figure 6.

CLMET outperforms CLM over CONUS with a smaller RMSE (0.31 versus 0.40 against MODIS,

0.19 versus 0.25 against FLUXNET-MTE). The improvement mainly results from reduction of the overestimation existing in CLM for SON and DJF. However, the model performance greatly varies with region. As indicated by the ET RMSE values, CLMET and CLM perform similarly over western CONUS, whereas CLMET improves the ET simulation over eastern CONUS no matter which reference data is used. Figure 7 compares the temporal evolution of the simulated ET in

CLM and CLMET against MODIS and FLUXNET-MTE ET over CONUS and four sub-regions.

It is evident that the bias correction method in CLMET is very effective in reducing overestimation (positive bias), but does not work as well in correcting the underestimation (negative bias) in water- limited regimes. The difference has to do with the specific ET regime, i.e. whether ET is limited by water or energy.  When an overestimated ET is overwritten with a lower value, the water on land is sufficient to support the reduced ET; in contrast, when an underestimated ET is overwritten with a higher value, the land surface model checks whether water storage in soil layer and vegetation canopy can sustain the elevated ET and further adjust if necessary to keep with the mass conservation equation. The extent to which ET can be increased is limited by the availability of water stored in soil layer and vegetation canopy. Therefore, in water-limited ET regimes, if ET is underestimated in CLM, the actual ET in CLMET after the water availability check can be substantially lower than the corrected ET fed into the model, which diminishes the effect of the bias correction algorithm under such circumstance.

In addition, the ET validation is also conducted at the site scale (Figures 8, 9, and 10).

Except for Port Peck and Wind River Crane stations in the northwest CONUS, for all other stations the monthly mean ET from CLMET agrees better with the observed ET than that from CLM

(Figure 8). The same statement holds for daily mean ET (Figures 9 and 10). Generally, CLM

overestimates ET as compared with station observations, and CLMET alleviates this overestimation, which is consistent with comparisons between the modelled ET and satellite-based

ET products.

4.2.2 Runoff

Using the runoff coefficient (the ratio of runoff to total precipitation) derived from GSCD

as the benchmark, we evaluate the model performance in CLM and CLMET in simulating runoff (Figure 11). The CONUS-averaged runoff coefficients in CLM and CLMET are 0.18 and 0.21, which are comparable to the GSCD-based runoff coefficient (0.22). However, CLM

underestimates runoff in most areas of CONUS due to an overestimation of ET. CLMET alleviates the underestimation by reducing ET therefore increasing the runoff, especially over eastern

CONUS. The relative bias of CLMET against GSCD is 1.1%, which is much smaller than the value in CLM (-9.2%). Table 4 shows the regional difference in runoff simulations in CLM and

CLMET. The improvement is greater over Eastern CONUS than Western CONUS, which is consistent with the improvement of ET simulations. The most striking improvement occurs in

Southeast CONUS, with the relative bias (RMSE) reduced from -24.7% (0.091) to -8.2% (0.06).

Because only the multi-year mean annual runoff coefficient is available for GSCD, we cannot examine the seasonal dependency of the model performance improvement.

The increase in runoff from CLM to CLMET is mainly due to the increase in subsurface runoff (not shown). The same value of the ET scaling factor within each grid cell are applied to three components of ET (interception loss, plant transpiration and soil evaporation) in this study.

Because interception loss accounts for a small portion of total ET, the absolute change of interception loss (decrease from CLM to CLMET over most areas) is much smaller compared with plant transpiration and soil evaporation (not shown). As a result, the increase in throughfall does not change much from CLM to CLMET, which leads to smaller increases in surface runoff. By contrast, plant transpiration and soil evaporation are more significantly reduced by CLMET, inducing wetter soil and therefore more subsurface runoff.

4.2.3 Soil moisture

As analyzed in Section 4.2.2, reductions in all three components of ET interception loss, plant transpiration, and soil evaporation from CLM to CLMET slow down moisture depletion from the soil. As a result, the water content in different soil layers increases with reduced ET. Figure 12

shows soil water at the surface and root-zone layers simulated by CLM and CLMET, and their differences in August. From CLM to CLMET, the changes over CONUS show an overwhelmingly increasing signal for both surface and root-zone soil moisture. The moisture increase in the top 0-

100 cm soil layer from CLM to CLMET in central CONUS is very evident, which may have significant implications in drought monitoring and assessment. For example, Central Great Plains experienced a severe drought in summer of 2012, and soil moisture derived from land surface models was used to evaluate the intensity of the drought event (Hoerling et al. 2014, Livneh and

Hoerling 2016). Unfortunately, land surface models tend to systematically overestimate drought (Milly and Dunne 2016, Ukkol et al. 2016). The more accurate estimates of ET and soil moisture resulting from the bias correction method in this study may prove useful for improving drought monitoring and assessment.

Due to the strong spatial heterogeneity of soil moisture and the lack of large-scale distributed data, the comparisons between observed soil moisture and modeled soil moisture from

CLM and CLMET are done based on the spatial averages across stations within each state and at the monthly scale during 2006-2010 for the top 0-10 cm and top 0-100 cm soil, respectively. The soil water increase from CLM to CLMET is more evident during SON and DJF, which is consistent with changes in ET that also features more decreases during SON and DJF. The soil in CLM shows dry biases over most of the examined states with the exception of soil moisture at the top 10 cm layer in Alabama and Illinois, and CLMET generally alleviate these dry biases. The RMSE values against the NASMD observations in CLMET is smaller than or at least the same as the RMSE

values in CLM. An exception exists for the top 0-10 cm layer in Alabama and Illinois where a wet bias is found in CLM. The soil water content difference between CLM and CLMET is larger for the 0-100 cm layer than the 0-10 cm layer, because plant transpiration, to which a large fraction of

ET and therefore a large fraction of ET bias correction are associated, primarily depletes moisture from the rooting zone which is deeper than 10 cm. As such, the improvement is more evident for the top 0-100 cm layer. For example, in Mississippi, the RMSE is reduced from 0.048 $m^3$ $m^{-3}$ in

CLM to 0.042 in CLMET at the top 0-10 cm layer, and from 0.07 to 0.06 $m^3$ $m^{-3}$ at the top 0-100

cm layer. The improvements in Alabama, Mississippi, Nebraska, and Oklahoma are summarized in Table 5.

**5 Summary and discussions**

In this study, we implemented the on-line bias correction approach proposed by Parr et al. (2015) to CLM4.5, and evaluated the effectiveness of the approach in reducing model biases over CONUS. The bias correction algorithm was calibrated using the GLEAM ET product combined with the default CLM4.5 output over the period of 1986-1995, and was validated over the period of 2000-2014 using both gridded and site-based ET datasets, the GSCD runoff product, and the NASMD soil moisture data. Results from all evaluation metrics indicate improved estimation of the terrestrial hydrological cycle across most of the model domain, with different degrees of improvement among the CONUS sub-regions.

Qualitatively, whether the Parr et al. (2015) ET bias correction approach improves the quantification of the hydrological cycle depends on whether ET is limited by water or energy *and* whether ET is underestimated or overestimated. The approach works well when/where ET is not limited by water availability; in water-limited regimes, the approach is effective in correcting the positive ET biases but does not work well if ET is underestimated. Quantitatively, the degree of the model improvement derived from this bias correction algorithm is highly related to whether the fundamental assumption of Parr et al. (2015) (on a time-invariant relationship characterizing the default model biases) holds or not. Although the scaling factors between observations and simulations do not change much from the calibration period to the validation period over most regions in most seasons, dramatic changes do exist in some areas. Differences in the scaling factors between the calibration and verification/application periods greatly influence the effectiveness of the bias correction method, with large differences causing the approach to be less effective leaving substantial biases in CLMET. Northeast CONUS during winter is an example of having a large bias in CLMET due to greater changes in the ET scaling factor from the calibration period to the verification period.

Another factor affecting the degree of the model improvement is whether the ET scaling is applied at all. As shown in Figure 2, we do not scale ET in some areas of Northwest CONUS

during the winter months due to the inconsistence in the ET sign (positive or negative) between

GLEAM and CLM. In these areas and season(s), ET in CLMET is not corrected at all. All these three factors (i.e., whether the scaling factor differs significantly between calibration and validation periods, whether ET is underestimated in water-limited regimes, and whether ET scaling is applied at all) influence the effectiveness of the bias correction approach, but one or two of them may dominate for a given region/season. For example, regardless of which product is used as the reference for comparison (Figures 3g, 5a4, 5b4), the approach reduces ET biases over the eastern

CONUS where the ET scaling is applied in most places/seasons and the scaling factor shows little difference between the calibration and validation periods. In contrast, in the north part of the

Midwest, some positive biases still remain in CLMET, as the ET scaling is not applied in winter months and the scaling factor differs quite much between these two periods. Over a portion of western CONUS, the bias correction approach is less effective due to the underestimation of ET

under a water-limited condition and large differences between calibration and validation periods in the scaling factor.

For a given grid cell and given month, the scaling factors for all three ET components, i.e., interception loss, plan transpiration, soil evaporation, are the same in this study, set to be the ratio of the remote sensing ET to the modeled ET. Since the GLEAM dataset contains values of three components besides the total ET, we conducted additional experiments in which the scaling factor for each ET component was estimated separately, using the ratio of each ET component from the

GLEAM product to the corresponding ET component from CLM during the same calibration period. However, results based on the component-specific scaling do not show further improvement, which is likely due to the inaccurate partitioning of ET into interception loss, plan transpiration, soil evaporation. Miralles et al. (2016) compared the ET partitioning for three widely used remote sensing-based ET products, and found that the contribution of each component to ET

is dramatically different among these three products. For instance, they found the percentage of global ET accounted for by soil evaporation ranges from 14% to 52%, and the ranges are even larger at the regional and local scales. Because the in-situ measurements of separate components of ET is very scarce, it is particularly challenging to validate the accuracy of the remote sensing- based estimates of the three ET components. These challenges led Miralles et al. (2016) to recommend against the use of any single product in partitioning ET.

The bias correction method evaluated in this study can effectively improve the estimates of surface fluxes and state variables in the absence of improved physical parameterizations in land surface models. It is applicable to not only historical simulations but also future predictions (Parr et al. 2015). It provides an alternative approach to, but would in no way replace, model improvement through better parameterization of physical processes. Development of better physical parameterizations has to be based on improved understanding of physical processes, more effective mathematical formulations, and higher quality surface type dataset, which requires a long-term commitment from the land surface modeling community. Model parameter calibration (e.g., tuning surface resistance) is another way to reduce model bias (Ren et al. 2016). However, the parameter space may contain nonphysical parameter subsets (Ray et al. 2015), which is especially an issue when model parameter tuning is used to offset unrelated model deficits. The method used in this study attempts to avoid such issues through improving the model performance without dealing with calibration of model physical parameters. ==However, results from this study==

==can provide useful guidance for physically-based land surface model development. As can be seen==

==from Figure 3g, the bias correction algorithm improves ET estimation over most of the U.S.,==

==indicating a strong potential for performance improvement that can be derived from improving the==

==physical parameterization of ET processes in the model. Over regions where the bias correction==

==approach does not improve the ET estimate (which are mostly places where ET is water-limited==

==while the model underestimates ET), parameterizations for other processes that influence soil==

==moisture (e.g., runoff generation, groundwater interactions) are the most likely cause for model==

==biases and should be the focus of physically-based model development effort.==

[revised manuscript text omitted]

Oleson, K. W., Lawrence, D. M., Bonan, G. B., Drewniak, B., Huang, M., Koven, C. D., Levis,

S., Li, F., Riley, W. J., Subin, Z. M., Swenson, S. C., Thornton, P. E., Bozbiyik, A., Fisher,

R. A., Kluzek, E., Lamarque, J.-F., Lawrence, P. J., Leung, L. R., Lipscomb, W., Muszala,

S., Ricciuto, D. M., Sacks, W. J., Sun, Y., Tang, J. Y., and Yang, Z.-L.: Technical Description of version 4.5 of the Community Land Model (CLM), NCAR Tech. Note, NCAR/TN-

503+STR, doi:10.5065/D6RR1W7M, 2013.

Parr, D., Wang, G., and Bjerklie, D.: Integrating Remote Sensing Data on Evapotranspiration and

Leaf Area Index with Hydrological Modeling: Impacts on Model Performance and Future

Predictions, J. Hydrometeorol., 16, 2086-2100, 2015.

Parr, D., Wang, G., and Fu, C.: Understanding Evapotranspiration Trends and their Driving

Mechanisms over the NLDAS Domain Based on Numerical Experiments Using CLM4.5,

Journal of Geophysical Research Atmospheres, 121, 7729-7745, 2016.

628 Ray, J., Z. Hou, M. Huang, K. Sargsyan, and L. Swiler: Bayesian calibration of the Community

629   Land Model using surrogates, SIAM/ASA Journal on Uncertainty Quantification, 199–233,

630   2015.

631 Reichle, R. H. and Koster, R. D.: Global assimilation of satellite surface soil moisture retrievals

632   into the NASA Catchment land surface model, Geophys. Res. Lett., 32, 177-202, 2005.

633 Ren, H., Z. Hou, M. Huang, J. Bao, Y. Sun, T. Tesfa, and R. Leung: Classification of hydrological

634   parameter sensitivity and evaluation of parameter transferability across 431 US MOPEX

635   basins, J. Hydrol., 536, 92–108, 2016.

636 Rodell, M., Houser, P. R., Jambor, U., Gottschalck, J. C., Mitchell, K., Meng, C. J., Arsenault, K.

637   R., Cosgrove, B. A., Radakovich, J., Bosilovich, M. G., Entin, J. K., Walker, J. P., Lohmann,

638   D., and Toll, D. L.: The Global Land Data Assimilation System, B. Am. Meteorol. Soc., 85,

639   381-394, 2004.

640 Sheffield, J. and Wood, E. F.: Characteristics of global and regional drought, 1950-2000: Analysis

641   of soil moisture data from off-line simulation of the terrestrial hydrologic cycle, Journal of

642   Geophysical Research Atmospheres, 112, D17115, 2007.

643 Spennemann, P. C. and Saulo, A. C.: An estimation of the land-atmosphere coupling strength in

644   South America using the Global Land Data Assimilation System, Int. J. Climatol., 35, 4151-

645   4166, 2015.

646 Swenson, S. C. and Lawrence, D. M.: A GRACE‐based assessment of interannual groundwater

647   dynamics in the Community Land Model, Water Resour. Res., 51, 8817-8833, 2015.

648 Syed, T. H., Famiglietti, J. S., Rodell, M., Chen, J., and Wilson, C. R.: Analysis of terrestrial water

649   storage changes from GRACE and GLDAS, Water Resour. Res., 44, 339-356, 2008.

Ukkola, A. M., Kauwe, M. G. D., Pitman, A. J., Best, M. J., Abramowitz, G., Haverd, V., Decker,

M., and Haughton, N.: Land surface models systematically overestimate the intensity, duration and magnitude of seasonal-scale evaporative droughts, Environ. Res. Lett., 11, 2016.

Wang, A., Zeng, X., and Guo, D.: Estimates of global surface hydrology and heat fluxes from the

Community Land Model (CLM4.5) with four atmospheric forcing datasets, J.

Hydrometeorol., 17, 2493-2510, 2016.

Xia, Y., Cosgrove, B. A., Mitchell, K. E., Peters Lidard, C. D., Ek, M. B., Kumar, S., Mocko, D., and Wei, H.: Basin‐scale assessment of the land surface energy budget in the National

Centers for Environmental Prediction operational and research NLDAS-2 systems, J.

Geophys. Res., 121, 196-220, 2016a.

Xia, Y., Ford, T. W., Wu, Y., Quiring, S. M., and Ek, M. B.: Automated Quality Control of In Situ

Soil Moisture from the North American Soil Moisture Database Using NLDAS-2 Products,

Journal of Applied Meteorology & Climatology, 54, 2015a.

Xia, Y., Hobbins, M. T., Mu, Q., & Ek, M. B.  Evaluation of NLDAS-2 evapotranspiration against tower flux site observations. Hydrological Processes, 29(7), 1757-1771, 2015b.

Xia, Y., Mitchell, K. E., Ek, M. B., Cosgrove, B., Sheffield, J., Luo, L., Alonge, C., Wei, H., Meng,

J., Livneh, B., Duan, Q., and Lohmann, D.: Continental-scale water and energy flux analysis and validation for North American Land Data Assimilation System project phase 2

(NLDAS‐2): 2. Validation of model‐simulated streamflow, J. Geophys. Res., 117, D3110,

2012a.

Xia, Y., Mitchell, K., Ek, M., Sheffield, J., Cosgrove, B., Wood, E., Luo, L., Alonge, C., Wei, H.,

Meng, J., Livneh, B., Lettenmaier, D., Koren, V., Duan, Q., Mo, K., Fan, Y., and Mocko, D.:

Continental-scale water and energy flux analysis and validation for the North American Land

Data Assimilation System project phase 2 (NLDAS-2): 1. Intercomparison and application of model products, Journal of Geophysical Research Atmospheres, 117, D3109, 2012b.

Xia, Y., Peters-Lidard, C. D., and Luo, L.: Basin-scale assessment of the land surface water budget in the National Centers for Environmental Prediction operational and research NLDAS-2

systems, J. Geophys. Res., 121, 196-220, 2016b.

Yin, J., Zhan, X., Zheng, Y., Liu, J., Fang, L., and Hain, C. R.: Enhancing Model Skill by

Assimilating SMOPS Blended Soil Moisture Product into Noah Land Surface Model, J.

Hydrometeorol., 16, 917-931, 2015.

Table 1 Spatial evaluations of simulated ET from two different types of runs (CLM and

CLMET) against GLEAM-derived ET over CONUS, Northwest (NW), Southwest (SW),

Northeast (NW), and Southeast (SW) annually and seasonally during the period 2000-2014.

March-April-May: MAM, June-July-August: JJA, September-October-November: SON,

December-January-February: DJF

| Season | Region | Bias (mm day$^{-1}$) | | Relative bias (%) | | RMSE (mm day$^{-1}$) | |
|---|---|---|---|---|---|---|---|
| | | CLM | CLMET | CLM | CLMET | CLM | CLMET |
| Annual | CONUS | 0.137 | -0.006 | 10.8 | -0.1 | 0.266 | 0.144 |
| | NW | 0.029 | -0.03 | 7.9 | 0.3 | 0.25 | 0.199 |
| | SW | 0.074 | -0.025 | 10.2 | -3.1 | 0.181 | 0.118 |
| | NE | 0.138 | -0.012 | 9.6 | -0.1 | 0.243 | 0.132 |
| | SE | 0.315 | 0.041 | 15.6 | 2.1 | 0.355 | 0.099 |
| MAM | CONUS | -0.081 | -0.062 | -5.8 | -3.3 | 0.351 | 0.227 |
| | NW | -0.138 | -0.074 | -6.7 | -2.7 | 0.326 | 0.244 |
| | SW | -0.211 | -0.122 | -17.9 | -9.3 | 0.318 | 0.206 |
| | NE | -0.191 | -0.078 | -8.3 | -2.8 | 0.429 | 0.293 |
| | SE | 0.19 | 0.023 | 8.9 | 1.5 | 0.346 | 0.165 |
| JJA | CONUS | 0.094 | -0.041 | 6.4 | -1.3 | 0.451 | 0.331 |
| | NW | -0.137 | -0.121 | -3.9 | -4.0 | 0.487 | 0.408 |
| | SW | 0.147 | -0.006 | 18.3 | -0.9 | 0.352 | 0.232 |
| | NE | 0.045 | -0.124 | 2.5 | -2.7 | 0.55 | 0.452 |
| | SE | 0.332 | 0.075 | 9.1 | 2.1 | 0.414 | 0.181 |
| SON | CONUS | 0.360 | 0.055 | 51 | 7.8 | 0.428 | 0.155 |
| | NW | 0.271 | 0.044 | 76.4 | 14.0 | 0.346 | 0.147 |
| | SW | 0.228 | 0.044 | 39.5 | 5.0 | 0.282 | 0.117 |
| | NE | 0.481 | 0.077 | 50.4 | 7.3 | 0.527 | 0.242 |
| | SE | 0.499 | 0.061 | 34.5 | 4.1 | 0.531 | 0.11 |
| DJF | CONUS | 0.182 | 0.009 | 77.7 | 18.9 | 0.265 | 0.115 |
| | NW | 0.114 | -0.013 | 104.2 | 28.8 | 0.252 | 0.122 |
| | SW | 0.132 | -0.014 | 42.3 | -1.9 | 0.182 | 0.056 |
| | NE | 0.239 | 0.077 | 146.4 | 65.3 | 0.334 | 0.199 |
| | SE | 0.24 | 0.004 | 49.5 | 2.7 | 0.292 | 0.072 |

Table 2. Similar to Table 1, but based on comparison with MODIS-derived ET during the period 2000-2011.

| Season | Region | Bias (mm day$^{-1}$) | | Relative bias (%) | | RMSE (mm day$^{-1}$) | |
|--------|--------|------|-------|------|-------|------|-------|
| | | CLM | CLMET | CLM | CLMET | CLM | CLMET |
| Annual | CONUS | 0.321 | 0.177 | 30.8 | 19.1 | 0.427 | 0.321 |
| | NW | 0.28 | 0.232 | 35.8 | 27.9 | 0.367 | 0.326 |
| | SW | 0.282 | 0.183 | 39.7 | 25.6 | 0.428 | 0.36 |
| | NE | 0.278 | 0.125 | 19.6 | 9.1 | 0.316 | 0.193 |
| | SE | 0.431 | 0.159 | 24.9 | 10.6 | 0.538 | 0.348 |
| MAM | CONUS | 0.514 | 0.533 | 50.1 | 55.8 | 0.631 | 0.635 |
| | NW | 0.564 | 0.628 | 67.2 | 74.5 | 0.636 | 0.687 |
| | SW | 0.345 | 0.438 | 45.9 | 61.8 | 0.538 | 0.599 |
| | NE | 0.547 | 0.655 | 51.7 | 61.9 | 0.58 | 0.675 |
| | SE | 0.596 | 0.436 | 34.6 | 25.8 | 0.735 | 0.578 |
| JJA | CONUS | 0.251 | 0.116 | 18.2 | 12.1 | 0.759 | 0.691 |
| | NW | 0.263 | 0.281 | 23.8 | 25.6 | 0.704 | 0.71 |
| | SW | 0.344 | 0.192 | 28.8 | 14.5 | 0.806 | 0.724 |
| | NE | 0.028 | -0.144 | 2.9 | -2.4 | 0.662 | 0.564 |
| | SE | 0.31 | 0.052 | 13.2 | 5.8 | 0.829 | 0.72 |
| SON | CONUS | 0.345 | 0.039 | 48.2 | 9.8 | 0.459 | 0.284 |
| | NW | 0.261 | 0.038 | 56.8 | 9.4 | 0.369 | 0.261 |
| | SW | 0.284 | 0.096 | 55.9 | 20.8 | 0.43 | 0.306 |
| | NE | 0.448 | 0.043 | 47.4 | 5.6 | 0.483 | 0.207 |
| | SE | 0.417 | -0.019 | 32.1 | 2.7 | 0.547 | 0.329 |
| DJF | CONUS | 0.181 | 0.025 | 82.2 | 28 | 0.383 | 0.276 |
| | NW | 0.043 | -0.049 | 77.6 | 40.4 | 0.385 | 0.365 |
| | SW | 0.156 | 0.007 | 70.5 | 19.4 | 0.292 | 0.191 |
| | NE | 0.091 | -0.051 | 96.7 | 14.8 | 0.344 | 0.214 |
| | SE | 0.403 | 0.169 | 87.5 | 33.6 | 0.474 | 0.281 |

Table 3. Similar to Table 1, but based on comparison with FLUXNET-MTE ET during the

                          period 2000-2011.

| Season | Region | Bias (mm day[-1]) | | Relative bias (%) | | RMSE (mm day[-1]) | |
|--------|--------|------|-------|------|-------|------|-------|
| | | CLM | CLMET | CLM | CLMET | CLM | CLMET |
| Annual | CONUS | 0.207 | 0.065 | 13.3 | 3.2 | 0.328 | 0.24 |
| | NW | 0.07 | 0.013 | 5.8 | 0.0 | 0.222 | 0.234 |
| | SW | 0.051 | -0.047 | 6.8 | -4.7 | 0.244 | 0.241 |
| | NE | 0.309 | 0.165 | 21.9 | 12.2 | 0.334 | 0.238 |
| | SE | 0.427 | 0.154 | 21.3 | 7.6 | 0.461 | 0.248 |
| MAM | CONUS | 0.27 | 0.292 | 15.8 | 19.5 | 0.418 | 0.399 |
| | NW | 0.266 | 0.33 | 22.4 | 28.0 | 0.349 | 0.401 |
| | SW | -0.042 | 0.051 | -7.3 | 2.5 | 0.298 | 0.301 |
| | NE | 0.288 | 0.401 | 21.6 | 30.4 | 0.338 | 0.435 |
| | SE | 0.561 | 0.4 | 26.4 | 18.5 | 0.6 | 0.448 |
| JJA | CONUS | 0.197 | 0.063 | 7.0 | 0.5 | 0.608 | 0.517 |
| | NW | -0.149 | -0.13 | -8.7 | -7.5 | 0.506 | 0.506 |
| | SW | 0.029 | -0.122 | 9.2 | -6.1 | 0.594 | 0.555 |
| | NE | 0.415 | 0.257 | 13.6 | 8.8 | 0.492 | 0.369 |
| | SE | 0.565 | 0.304 | 16.9 | 9.4 | 0.779 | 0.585 |
| SON | CONUS | 0.216 | -0.088 | 20.3 | -9.4 | 0.353 | 0.294 |
| | NW | 0.072 | -0.151 | 9.2 | -22.8 | 0.224 | 0.286 |
| | SW | 0.132 | -0.055 | 21.1 | -5.2 | 0.311 | 0.277 |
| | NE | 0.356 | -0.034 | 33.7 | -1.1 | 0.473 | 0.385 |
| | SE | 0.346 | -0.091 | 21.2 | -5.4 | 0.396 | 0.23 |
| DJF | CONUS | 0.149 | -0.004 | 40.1 | -1 | 0.268 | 0.189 |
| | NW | 0.104 | 0.014 | 27 | -4.9 | 0.279 | 0.26 |
| | SW | 0.086 | -0.063 | 20.9 | -14.4 | 0.17 | 0.129 |
| | NE | 0.176 | 0.037 | 78.5 | 19.2 | 0.329 | 0.208 |
| | SE | 0.236 | 0.002 | 42.8 | 0.8 | 0.282 | 0.129 |

Table 4 Statistics of simulated annual runoff coefficient (ratio of runoff to total precipitation)

against GSCD observations over CONUS, Northwest (NW), Southwest (SW), Northeast (NW), and Southeast (SW) during the period 2000-2014.

|           | Bias   |        | Relative bias (%) |        | RMSE  |       |
|-----------|--------|--------|-------------------|--------|-------|-------|
|           | CLM    | CLMET  | CLM               | CLMET  | CLM   | CLMET |
| CONUS     | -0.053 | -0.027 | -18.5             | -6.7   | 0.198 | 0.192 |
| Northwest | -0.046 | -0.036 | -13.5             | -5.6   | 0.146 | 0.144 |
| Southwest | -0.026 | -0.019 | -19.9             | -11.4  | 0.373 | 0.373 |
| Northeast | -0.06  | -0.022 | -15.7             | -1.5   | 0.108 | 0.092 |
| Southeast | -0.074 | -0.026 | -24.7             | -8.2   | 0.091 | 0.06  |

Table 5 Root mean square error (RMSE) values of monthly volumetric soil moisture ($m^{-3}m^{-3}$)

simulated by CLM and CLMET relative to the quality-controlled NASMD for the top 0-10 cm soil layer and for the top 0-100 cm soil layer over Alabama, Illinois, Mississippi, Nebraska, and

                             Oklahoma.

|  | top 0-10 cm soil water content | | top 0-10 cm soil water content | |
| --- | --- | --- | --- | --- |
|  | CLM | CLMET | CLM | CLMET |
| Alabama | 0.044 | 0.048 | 0.027 | 0.020 |
| Illinois | 0.019 | 0.021 | 0.038 | 0.034 |
| Mississippi | 0.048 | 0.042 | 0.070 | 0.060 |
| Nebraska | 0.014 | 0.014 | 0.032 | 0.025 |
| Oklahoma | 0.050 | 0.045 | 0.039 | 0.032 |

[Figure]

Figure 1 a) Mean annual (1980-2015) precipitation in mm over conterminous USA

(CONUS). NW, SW, NE, and SE represent Northwest, Southwest, Northeast, and Southeast

CONUS, respectively. The black circles represent sites of in-situ soil moisture observations in

Alabama, Illinois, Mississippi, Nebraska, and Oklahoma. b) Locations of the 16 AmeriFlux stations with vegetation types.

[Figure]

Figure 2 Scaling factor as the ratio of the CLM simulated ET to the GLEAM ET for each month during 1986-1995. The numbers in titles are CONUS-averaged values, and the numbers of within figures are area-averaged values for each of four sub regions (NW, SW, NE, and SE). The areas with negative scaling factors are masked out.

[Figure]

Figure 3 Mean annual ET from a) GLEAM, b) CLM, and c) CLMET, the relative difference between d) CLMET and CLM, e) CLM and GLEAM, f) CLMET and GLEAM, and g) the difference between absolute value of e) and absolute value of f) during the period 2000-2014.

Numbers in titles are CONUS-averaged values.

[Figure]

Figure 4 a) Relative bias (RB) for CLM (RB_CLM), b) RB for CLMET (RB_CLMET) during the period 2000-2014, c) difference in scaling factor $f_{ET}$ between the period 1986-1995 and the period 2000-2014 ($f_{ET}(86)$- $f_{ET}(00)$), and d) scatter plots of $f_{ET}(86)$- $f_{ET}(00)$ versus RB_CLMET in 1)

January (Jan), 2) April (Apr), 3) July (Jul), and 4) November (Nov).

[Figure]

Figure 5 Mean annual ET from a1) MODIS, b1) FLUXNET-MTE, the relative differences between a2) CLM and MODIS, b2) CLM and FLUXNET-MTE, a3) CLMET and MODIS, and b3) CLMET and FLUXNET-MTE, and the differences between a4) absolute value of a2 and absolute value of a3, and b4) absolute value of b2 and absolute value of b3 during the period 2000-2011. Numbers in titles are CONUS-averaged values.

[Figure]

Figure 6 Seasonal cycles of ET from MODIS, FLUXNET-MTE, CLM, and CLMET over a)

CONUS, b) Northwest, c) Southwest, d) Northeast, and e) Southeast during the period 2000-

             2011.

[Figure]

Figure 7 Time series of ET difference between model (CLM or CLMET) and reference data (MODIS or FLUXNET-MTE) over a) CONUS, b) Northwest, c) Southwest, d) Northeast, and e)

          Southeast during the period 2000-2011.

[Figure]

Figure 8 Monthly mean latent heat fluxes from CLM, CLMET and observations at 16 flux tower sites. RMSE$_{CLM}$ and RMSE$_{CLMET}$ represent the root mean square error against observations for

CLM and CLMET, respectively. Note that the CLM and CLMET simulations are driven with meteorological forcings at the grid cell level (as opposed to site-specific forcing).

[Figure]

Figure 9 Daily mean latent heat fluxes from CLM and CLMET grids and station observations at

ARM SGP Burn, Audubon Grassland, Bondville, Donaldson, Flagstaff Forest, Fort Dix, Fort

Peck, and Little Prospect. $RMSE_{CLM}$ and $RMSE_{CLMET}$ represent the root mean square error against observations for CLM and CLMET, respectively.

[Figure]

Figure 10 Daily mean latent heat fluxes from CLM and CLMET grids and station observations at

Mead Rainfed, Metolius Pine, Missouri Ozark, Morgan Forest, Sylvania Wilderness, Tonzi

Ranch, Walnut River, and Wind River Crane. RMSE$_{CLM}$ and RMSE$_{CLMET}$ represent the root mean square error against observations for CLM and CLMET, respectively.

[Figure]

794  Figure 11 Mean annual runoff coefficient (the ratio runoff to total precipitation) from a) Global

795    Streamflow Characteristics Dataset (GSCD), b) CLM, and c) CLMET, and the relative

796    differences between d) CLM and GSCD, e) CLMET and GSCD, and f) CLMET and CLM

797 during the period 2000-2014. Runoff coefficient less than 0.02 is blanked out. Numbers in titles

798       are CONUS-averaged values.

[Figure]

Figure 12 Simulated soil moisture (mm) in the top a) 0-10 cm and b) 0-100 layers in August from 1) CLM and 2) CLMET, 3) their differences, and 4) their relative differences during the period 2000-2014.